# Optimal Task Order for Continual Learning of Multiple Tasks

Ziyan Li [1]   Naoki Hiratani [2]

## Abstract

Continual learning of multiple tasks remains a major challenge for neural networks. Here, we investigate how task order influences continual learning and propose a strategy for optimizing it. Leveraging a linear teacher-student model with latent factors, we derive an analytical expression relating task similarity and ordering to learning performance. Our analysis reveals two principles that hold under a wide parameter range: (1) tasks should be arranged from the least representative to the most typical, and (2) adjacent tasks should be dissimilar. We validate these rules on both synthetic data and real-world image classification datasets (Fashion-MNIST, CIFAR-10, CIFAR-100), demonstrating consistent performance improvements in both multilayer perceptrons and convolutional neural networks. Our work thus presents a generalizable framework for task-order optimization in task-incremental continual learning.

## 1. Introduction

The ability to learn multiple tasks continuously is a hallmark of general intelligence. However, deep neural networks and its applications, including large language models, struggle with continual learning and often suffer from catastrophic forgetting of previously acquired knowledge (McCloskey & Cohen, 1989; French, 1999; Hadsell et al., 2020; Luo et al., 2023). Although extensive work has been done to identify when forgetting is most prevalent (Ramasesh et al.; Lee et al., 2021) and how to mitigate it (French, 1991; Robins, 1995; Kirkpatrick et al., 2017; Shin et al., 2017; Serra et al., 2018; Rolnick et al., 2019), it remains unclear how to prevent forgetting while simultaneously promoting knowledge transfer across tasks (Ke et al., 2020; Lin et al., 2022; Ke

[1]Department of Physics, Washington University in St Louis, St Louis, USA [2]Department of Neuroscience, Washington University in St Louis, St Louis, USA. Correspondence to: Naoki Hiratani <hiratani@wustl.edu>.

*Proceedings of the 42$^{nd}$ International Conference on Machine Learning*, Vancouver, Canada. PMLR 267, 2025. Copyright 2025 by the author(s).

et al.; Kontogianni et al., 2024).

One important yet relatively underexplored aspect of continual learning is task-order dependence. Previous work has revealed that the order in which tasks are presented can significantly influence continual learning performance and also explored various approaches to optimize task order (Lad et al., 2009; Pentina et al., 2015; Guo et al., 2018; Bell & Lawrence, 2022; Lin et al., 2023; Singh et al., 2023). However, we still lack clear understanding on how ordering of tasks influences the learning performance and how to order tasks to achieve optimal performance. Figure 1 illustrates this problem using a continual binary image classification example, where a neural network is trained on three tasks: $A$ (Cat vs. Ship), $B$ (Frog vs. Truck), and $C$ (Horse vs. Deer). Learning one task can influence performance on the others in a complex manner (Figs. 1a and 1b). Consequently, in this example, the $C{\rightarrow}B{\rightarrow}A$ task order achieves a higher average performance after training than the $A{\rightarrow}B{\rightarrow}C$ order (Fig. 1c). The goal of this work is to understand this task-order dependence in continual learning.

Task-order optimization requires some amount of knowledge on all tasks beforehand, making it infeasible in a strictly online learning setting. Nevertheless, it remains highly relevant for many learning problems. One scenario is when data acquisition and training need to be conducted in parallel, which may occur in the training of self-driving algorithms (Verwimp et al., 2023) or medical image analysis (Kumari et al., 2023). In this setting, it is beneficial to first collect a small pilot dataset across all underlying tasks and then determine the optimal order of data acquisition and training to maximize knowledge transfer while minimizing forgetting across tasks. To demonstrate the potential applicability of our theory to this problem, we provide numerical evidence that, in continual visual recognition benchmarks, an optimal task order estimated from just 1% of the data significantly outperforms a random task order.

Moreover, in robotics applications (Lesort et al., 2020; Ibarz et al., 2021), switching between tasks often involves physically rearranging objects around the robot, which is both time-consuming and labor-intensive. As a result, switching tasks on a trial-by-trial basis is often infeasible, necessitating block-wise training. In this scenario, optimizing task order could help maximize average performance across all tasks.

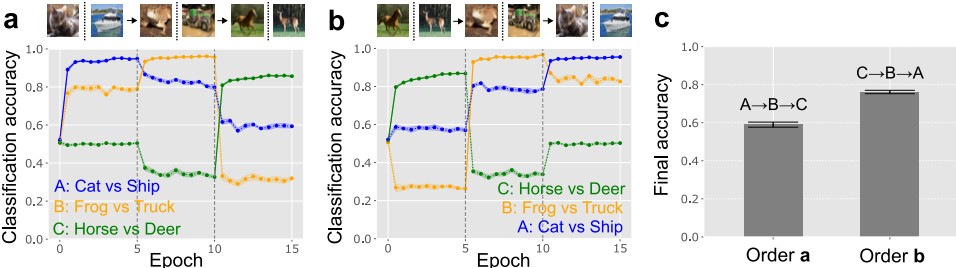

*Figure 1.* Schematic figure of the task-order dependence. **a, b)** Continual learning of binary classification with two different task orders. **c)** Average test accuracy on the three classification tasks at the end of learning under task orders depicted in panels **a** and **b**. Error bars represent the standard error of mean over 10 random seeds.

Similar constraints arise in designing of machine-learning-based teaching curricula for schools or professional training where learners need to study multiple subjects sequentially (Rafferty et al., 2016; Singh et al., 2023). Furthermore, even in a more traditional continual learning task, if the current task creates unfavorable conditions, systems can postpone its learning to a more suitable time. Understanding how the order of tasks impacts learning can also serve as a tool for predicting the difficulty of online learning given a data stream. Lastly, large language models are typically trained in an online fashion because the size of the training corpus is so vast that multiple epochs of training over the entire dataset is infeasible (Hoffmann et al., 2022; Chowdhery et al., 2023). In such cases, how the corpus is organized for training can significantly impact learning efficiency, supporting the importance of optimizing task/corpus order.

To explore the basic principle of task order optimization, here we analyze the task-order dependence of continual learning using a linear teacher-student model with latent factors. First, we derive an analytical expression for the average error after continual learning as a function of task similarity for an arbitrary number of tasks. Our theory shows that this error inevitably depends on the task order because it is a function of the upper-triangular components of the task similarity matrix, rather than of the entire matrix.

We then investigate how the similarity between tasks, when placed in various positions within the task order, affects the overall error. Through linear perturbation analysis, we find that the task-order effect decomposes into two factors. The first is absolute order dependence: similarity between two tasks influences the error differently depending on whether these tasks appear near the beginning or near the end of the sequence. We demonstrate that when tasks are on average positively correlated, the least representative tasks should be learned first, while the most typical task should be learned last (periphery-to-core rule). The second factor is relative order dependence: the effect of task similarity on the error differs depending on whether two tasks are adjacent in the sequence or far apart. We show that a task order maximizing

the path length in the task dissimilarity graph outperforms one that minimizes this path length (max-path rule), consistent with previous empirical observations (Bell & Lawrence, 2022).

We illustrate these two rules by applying them to tasks with simple similarity structures forming chain, ring, and tree graphs, revealing the presence of non-trivial task orders that robustly achieve the optimal learning performance, given a graph structure. Moreover, we apply these rules to continual image classification tasks using the Fashion-MNIST, CIFAR-10, and CIFAR-100 datasets. We estimate task similarity by measuring the zero-shot transfer performance between tasks, and then implement the task-ordering rules based on these estimates. Our results show that both the periphery-to-core rule and the max-path rule hold robustly in both multilayer perceptrons and convolutional neural networks. Moreover, using $\sim 1\%$ of the data for the task similarity and order estimation was sufficient to achieve a significant improvement over random ordering. This work thus provides a simple and generalizable theory to task-order optimization in task-incremental continual learning.

## 2. Related Work

The effects of curriculum learning have been extensively studied in the reinforcement learning (RL) literature (Elman, 1993; Krueger & Dayan, 2009; Narvekar et al., 2020). However, these studies primarily focus on learning a single challenging task by sequentially training on simpler tasks, leaving open the question of how to design a curriculum for learning multiple tasks of similar difficulty. A limited number of works have explored task-order optimization for continual/lifelong learning across multiple tasks by contrast.

Lad et al. (2009) demonstrated that ordering tasks based on pairwise order preferences can lead to better classification performance compared to random task ordering. More recently, Bell & Lawrence (2022) investigated task-order optimization by examining Hamiltonian paths on a task dissimilarity graph (see Sec. 4 for details). They hypothe-

sized that the shortest Hamiltonian path would be optimal but instead found that the longest Hamiltonian path significantly outperformed both random task ordering and the shortest path in continual image classification tasks. Our work provides analytical insights into when and why this is the case. Lin et al. (2023) analyzed generalization error and task-order optimization in continual learning for linear regression. Our work advances this theoretical framework in several important ways. First, we introduce a latent structure model for considering the effect of input similarity and reveal how tasks' relative positions—not just their absolute positions as in Lin et al. (2023)'s Equation 10—influence the model's final performance. We validate this theoretical finding through experiments on both synthetic data and image classification tasks. Furthermore, we extend beyond the synthetic task settings of Lin et al. (2023) by demonstrating these effects in a general continual learning framework using data-driven similarity estimation. Task-order effects on continual learning have also been analyzed in (Pentina et al., 2015; Evron et al., 2023; Singh et al., 2023).

The linear teacher-student model used in this work is a widely adopted framework for analyzing the average properties of neural networks by explicitly modeling the data generation process through a teacher model (Gardner & Derrida, 1989; Zdeborová & Krzakala, 2016; Bahri et al., 2020). Due to their analytical tractability, these models have offered deep insights into various aspects of statistical learning problems, including generalization (Seung et al., 1992; Advani et al., 2020), learning dynamics (Saad & Solla, 1995; Werfel et al., 2003; Saxe et al., 2014), and representation learning (Saxe et al., 2019; Tian et al., 2021). Many studies have also applied this framework to explore various aspects of continual learning (Asanuma et al., 2021; Lee et al., 2021; Evron et al., 2022; Goldfarb & Hand, 2023; Li et al., 2023; Lin et al., 2023; Goldfarb et al.; Hiratani; Mori et al.).

## 3. Task-order Dependence

### 3.1. Model Setting

Let us consider a sequence of $P$ tasks, where the inputs $\boldsymbol{x} \in \mathbb{R}^{N_x}$ and the target output $\boldsymbol{y}^* \in \mathbb{R}^{N_y}$ of the $\mu$-th task is generated by

$$\boldsymbol{s} \sim \mathcal{N}(0, I), \quad \boldsymbol{x} = A_\mu \boldsymbol{s}, \quad \boldsymbol{y}^* = B_\mu \boldsymbol{s}. \quad (1)$$

Here, $\boldsymbol{s} \in \mathbb{R}^{N_s}$ is the latent factor that underlies both $\boldsymbol{x}$ and $\boldsymbol{y}^*$, $I$ is the identity matrix, and $A_\mu \in \mathbb{R}^{N_x \times N_s}$ and $B_\mu \in \mathbb{R}^{N_y \times N_s}$ are the mixing matrices that generate the input $\boldsymbol{x}$ and the target $\boldsymbol{y}^*$ from the latent $\boldsymbol{s}$ (Fig. 2a). Below we focus on $N_x \gg N_s$ regime. The introduction of this low-dimensional latent factor $\boldsymbol{s}$ is motivated by the presence of low-dimensional latent structures in many real-world datasets (Yu et al., 2017; Cohen et al., 2020).

We sample elements of $\{A_\mu, B_\mu\}_{\mu=1}^P$ from a correlated Gaussian distribution. Denoting a vector consists of the $(i, j)$-th elements of $A_1, .., A_P$ by $\boldsymbol{a}_{ij} \equiv [A_{1,ij}, A_{2,ij}, ...., A_{P,ij}]^T$, we sample $\boldsymbol{a}_{ij} \in \mathbb{R}^P$ from

$$\boldsymbol{a}_{ij} \sim \mathcal{N}\left(\boldsymbol{0}, \tfrac{1}{N_s} C^{in}\right) \quad (2)$$

where $C^{in}$ is a $P \times P$ matrices that specify input correlation between tasks. Similarly, we sample the $(i, j)$-th elements of $B_1, .., B_P$, $\boldsymbol{b}_{ij} \equiv [B_{1,ij}, B_{2,ij}, ...., B_{P,ij}]^T$, by $\boldsymbol{b}_{ij} \sim \mathcal{N}(\boldsymbol{0}, \tfrac{1}{N_s} C^{out})$. Note that here correlation is introduced across tasks in an element-wise manner while keeping elements of each mixing matrix independent (i.e. $\left\langle A_{ij}^\mu A_{kl}^\nu \right\rangle_A = \delta_{ik} \delta_{jl} \frac{C_{\mu\nu}^{in}}{N_s}$), where $\delta_{ik}$ represents the Kronecker delta. Here we generate the model from the task similarity matrices $\{C^{in}, C^{out}\}$ because previous work suggests the crucial impact of task similarity on continual learning (Ramasesh et al.; Lee et al., 2021). In section 5, we consider the estimation of task similarity from datasets to ensure the applicability of our framework.

Let us consider the training of a linear network, $\boldsymbol{y} = W\boldsymbol{x}$, in this set of $P$ tasks. We evaluate the performance of the network for the $\mu$-th task using the mean squared error:

$$\epsilon_\mu[W] \equiv \left\langle \|\boldsymbol{y}^* - \boldsymbol{y}\|^2 \right\rangle_{\boldsymbol{s}} = \|B_\mu - W A_\mu\|_F^2, \quad (3)$$

where $\langle \cdot \rangle_{\boldsymbol{s}}$ represents expectation over latent $\boldsymbol{s} \sim \mathcal{N}(\boldsymbol{0}, I)$. In a task-incremental continual learning task (Van de Ven & Tolias, 2019), we are mainly concerned with minimizing the total error on all tasks after learning all tasks. Denoting the network parameter after learning of the last, $P$-th, task by $W_P$, the final error is defined by

$$\epsilon_f \equiv \frac{1}{N_y} \sum_{\mu=1}^P \epsilon_\mu[W_P]. \quad (4)$$

Below, we take the expectation over randomly generated mixing matrices $\{A_\mu, B_\mu\}_{\mu=1}^P$ and derive the average final error $\bar{\epsilon}_f \equiv \langle \epsilon_f \rangle_{\{A_\mu, B_\mu\}}$ as a function of the input and output correlation matrices $C^{in}$ and $C^{out}$. Subsequently, we analyze how the task order influences $\bar{\epsilon}_f$ and how to optimize the order.

### 3.2. Analysis of the Final Error $\epsilon_f$

We consider task incremental continual learning where $P$ tasks are learned in sequence one by one. Let us denote the weight after training on the $(\mu - 1)$-th task as $W_{\mu-1}$. Considering learning of the $\mu$-th task from $W = W_{\mu-1}$ using gradient descent on task-specific loss $\epsilon_\mu[W]$, the weight after training follows (see Appendix A.2)

$$W_\mu = W_{\mu-1} \left(I - U_\mu U_\mu^T\right) + B_\mu A_\mu^+, \quad (5)$$

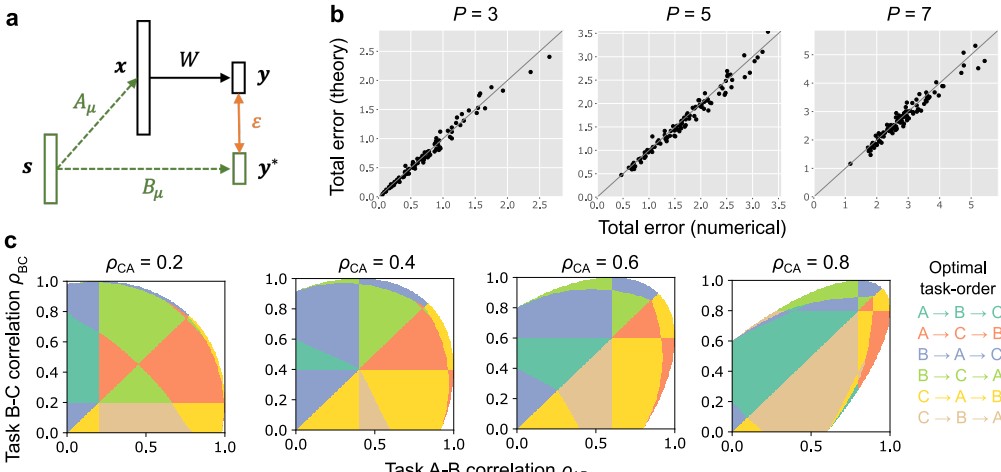

*Figure 2.* **a)** Schematic of the teacher-student model. **b)** Comparison between the analytical and numerical evaluations of the error $\epsilon_f$ under various number of tasks. Each point represents the errors under a randomly sampled task similarity matrices $(C^{in}, C^{out})$ (see Appendix C for implementation details). **c)** Optimal task order for three task learning. In the white regions, $C^{in}$ is not a positive-definite matrix, hence the tasks are not well-defined.

where $U_\mu$ is defined by singular value decomposition (SVD) of $A_\mu$, $A_\mu = U_\mu \Lambda_\mu V_\mu^T$, and $A^+$ is the pseudo-inverse of $A$. Applying it recursively while assuming that $W$ is initialized as a zero matrix prior to the first task, we have

$$W_\mu = \sum_{\nu=1}^{\mu} (B_\nu A_\nu^+) \prod_{\rho=\nu+1}^{\mu} (I - U_\rho U_\rho^T). \quad (6)$$

If $N_x \gg N_s$, pseudo-inverse $A_\mu^+$ is approximated by a scaled transpose $\gamma A_\mu^T$, and $U_\mu U_\mu^T$ approximately follows $U_\mu U_\mu^T \approx \gamma A_\mu A_\mu^T$ with $\gamma = \frac{N_s}{N_x}$ (see Appendix A.4). Thus, under $\frac{N_s}{N_x} \ll 1$, we have

$$W_\mu \approx \gamma \sum_{\nu=1}^{\mu} (B_\nu A_\nu^T) \prod_{\rho=\nu+1}^{\mu} (I - \gamma A_\rho A_\rho^T). \quad (7)$$

Under this approximation, there exists a simple expression of the final error as below (see Appendix A.3 for the proof).

**Theorem 3.1.** *At $\frac{N_s}{N_x} \to 0$ limit, the final error asymptotically satisfies*

$$\bar{\epsilon}_f = \left\| (C^{out})^{1/2} \left( I - (I + C^{in,U})^{-1} C^{in} \right) \right\|_F^2, \quad (8)$$

*where $C^{in,U}$ is the strictly upper-triangle matrix generated from the input correlation matrix $C^{in}$ (see eq. 34).*

Importantly, the dependence on the upper-triangular components in Eq. 3.1 implies that $\bar{\epsilon}_f$ is not permutation-invariance, and thus depends on the task-order.

### 3.3. Numerical Evaluation

To check this analytical result, in Fig. 2b, we compared $\bar{\epsilon}_f$ estimated from Eq. 8 with its numerical estimation through

learning via gradient descent, under various choices of the number of tasks $P$ and task correlation matrices $C^{in}$ and $C^{out}$ (each point in Fig. 2b represents the errors under one randomly sampled $\{C^{in}, C^{out}\}$ pair). This result indicates that our simple analytical expression robustly captures the performance of continual learning in a linear teacher-student model under arbitrary task similarity and the number of tasks.

To explore how task order influences the continual learning performance, we next calculated the optimal task order of three tasks under various input correlation $C^{in}$ using Eq. 8 (Fig. 2c). Here, we set the output correlation $C_{\mu\nu}^{out} = 1$ for all task pairs $(\mu, \nu)$ for simplicity, and parameterized the input correlation between tasks A, B, and C by

$$C^{in} = \begin{pmatrix} 1 & \rho_{AB} & \rho_{CA} \\ \rho_{AB} & 1 & \rho_{BC} \\ \rho_{CA} & \rho_{BC} & 1 \end{pmatrix}. \quad (9)$$

Here, tasks A, B, and C are linear regression tasks with partial overlap in the input domain. If $\rho_{AB} = 1$, the input subspace for tasks A and B are the same, while they are independent if $\rho_{AB} = 0$. Figure 2c revealed that the optimal task order depends on the combination of task similarity $(\rho_{AB}, \rho_{BC}, \rho_{CA})$ in a rich and complex manner. Some of the phase shifts represent trivial mirror symmetry (e.g., $x = y$ line), but many of them are non-trivial. To further gain insights into this complex task-order dependence, below, we consider the linear perturbation limit.

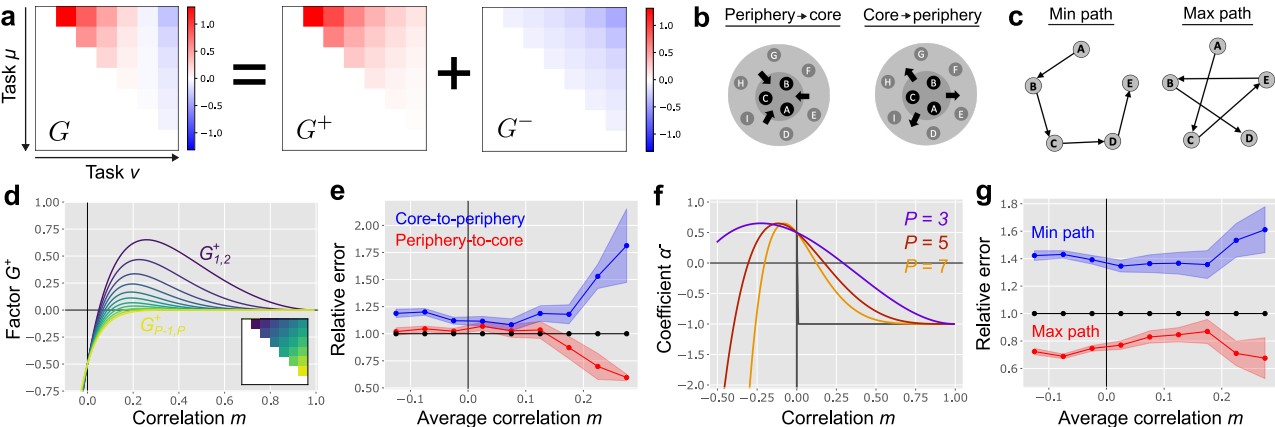

*Figure 3.* Linear perturbation analysis of task order dependence. **a)** Decomposition of the contribution of task similarity to the final error under $\rho_o = 1, m = 0.3$. **b)** Schematic representations of periphery-to-core ordering and core-to-periphery ordering. Circles A, B, ..., I represent tasks and their spatial positions represent similarity between tasks. Here, tasks A-C are central whereas tasks D-I are periphery. **c)** Schematic of minimum and maximum paths on a task dissimilarity graph. **d)** $G_{\mu\nu}^+$ as a function of $m$ under $P = 7$. Colors indicate the indices such that the purple line on the top corresponds to $G_{1,2}^+$, while the yellow line at the bottom corresponds to $G_{6,7}^+$. **e)** Relative error of core-to-periphery and periphery-to-core rules under various average task correlation $m$ at $P = 7$. Error bars represent the standard error over random seeds (see Appendix C for details). **f)** Coefficient $\alpha^-$ under $P = 3, 5, 7$. Dark gray line corresponds to $\alpha^-$ at $P \to \infty$ limit. **g)** Relative error of min-path and max-path task orders under $P = 7$. Here we took average over two task orders that follows the minimum pathway to estimate the error of min-path. The error of max-path was estimated in the same manner.

## 4. Task-order Optimization

### 4.1. Linear Perturbation Theory

Theorem 3.1 revealed a simple relationship between the task similarity and the final error of continual learning, but it remains unclear how to optimize the task order for continual learning. To gain insight into this question, we next add a small perturbation to the input similarity matrix and examine how the change in the similarity between various task pairs modifies the error. We parameterize the input correlation matrix by a combination of a constant factor and a small perturbation.

$$C_{\mu\nu}^{in} = \begin{cases} 1 & (\text{if} \quad \mu = \nu) \\ m + \delta M_{\mu\nu} & (\text{otherwise}). \end{cases} \quad (10)$$

Here, we set the constant factor $m$ to be the same across all tasks for the analytical tractability of the matrix inversion, and perturbation $\delta M$ is constrained to the ones that keep $C^{in}$ to a correlation matrix. Similarly, we restricted the target output correlation matrix to be $C_{\mu\nu}^{out} = \rho_o$ for all non-diagonal components. In this setting, the error has the following decomposition.

**Theorem 4.1.** *Let us suppose that all elements of matrix $\delta M$ satisfies, $|\delta M_{\mu\nu}| < \delta_m$, where $\delta_m$ is a positive constant.*

*Then, the final error is written as below:*

$$\bar{\epsilon}_f[C^{in}, C^{out}] = \bar{\epsilon}_f[m, \rho_o] + \sum_{\mu=1}^{P} \sum_{\nu=\mu+1}^{P} G_{\mu\nu} \delta M_{\mu\nu} + \mathcal{O}(\delta_m^2),$$

(11)

*where $\bar{\epsilon}_f[m, \rho_o]$ is the error in the absence of perturbation, and $G_{\mu\nu}$ is a function of $m$, $\rho_o$, and $P$ (see Eqs. 57 in Appendix). At $\rho_o = 1$, $G_{\mu\nu}$ has a following simple expression:*

$$G_{\mu\nu} = G_{\mu\nu}^+ + G_{\mu\nu}^-, \quad (12a)$$

$$G_{\mu\nu}^+ \equiv -(1-m)^{P+\mu-1} - (1-m)^{P+\nu-1}$$
$$+ \tfrac{3-m}{2-m}(1-m)^{\mu+\nu-1}, \quad (12b)$$

$$G_{\mu\nu}^- \equiv -\left(1 - (1-m)^P \left(\tfrac{mP}{1-m} + \tfrac{3-m}{2-m}\right)\right)(1-m)^{P-(\nu-\mu)}. \quad (12c)$$

Note that, $P \times P$ matrix $G$ specifies the contribution of $(\mu, \nu)$-th task similarity to the final error. The proof of the theorem is provided in Appendix B.2. Fig. 3a describes an example of $G_{\mu\nu}$ (here $P = 7$ and $m = 0.3$). In this case, $G_{12}$ is positive while $G_{17}$ is negative, meaning that if you increase the similarity between the first and the second tasks while keeping the rest the same, the total error $\bar{\epsilon}_f$ goes up, but if you increase the similarity between the first and the last tasks, the error instead decreases. To understand this task order dependence, we next analyze $G^+$ and $G^-$ separately.

## 4.2. Impact of Task Typicality

Let us first consider the contribution of $G_{\mu\nu}^+$ term. Denoting $\alpha^+ \equiv \frac{2-m}{3-m}(1-m)^P$, $G_{\mu\nu}^+$ is rewritten as

$$G_{\mu\nu}^+ = \frac{3-m}{(2-m)(1-m)}\left((1-m)^\mu - \alpha^+\right)\left((1-m)^\nu - \alpha^+\right) - \alpha^+(1-m)^{P-1}, \tag{13}$$

If $1 > m > 0$, $\frac{3-m}{(2-m)(1-m)} > 0$ and $(1-m)^\mu \geq (1-m)^P > \alpha^+$ for $\mu = 1, 2, ..., P$. Therefore, $G_{\mu\nu}^+$ is a monotonically decreasing function of both $\mu$ and $\nu$ under $1 > m > 0$ (Fig. 3d). This means that, to minimize the error contributed from $G_{\mu\nu}^+$, $\delta\epsilon_f^+ \equiv \sum_{\mu,\nu} G_{\mu\nu}^+ \delta M_{\mu\nu}$, the tasks should be ordered in a way that the residual similarity $\delta M_{\mu\nu}$ takes a small (preferably negative) value for early task pairs and a large value for later task pairs. In other words, earlier pairs should be relatively dissimilar to each other, while later pairs should be more similar.

One heuristic way to achieve this task order is to put the most atypical task at the beginning and the most typical one at the end. Denoting the relative typicality of the task by $\delta t_\mu = \sum_{\nu \neq \mu} \delta M_{\nu\mu}$, if we arrange tasks as $\delta t_1 \leq \delta t_2 \leq ... \leq \delta t_P$, on average, earlier pairs are dissimilar to each other while the latter ones are similar. Below, we denote this ordering as a periphery-to-core rule, as less representative periphery tasks are learned first and more central core tasks are learned later under this principle (Fig. 4b). Under a randomly generated input correlation matrix $C^{in}$, periphery-to-core task order robustly outperformed both random and core-to-periphery order, when the average correlation is large positive value (red vs black and blue line in Fig. 3e). This was not the case when the average correlation is a small positive value potentially due to contribution from $G^-$ factor. Note that, a similar rule was derived by Lin et al. (2023) based on their analysis of linear regression model, where they proved that when there is one outlier task, the outlier task should be learned in the first half of the task sequence.

## 4.3. Impact of Hamiltonian Path Length

Let us next focus on $G_{\mu\nu}^-$ term that governs the contribution of the relative distance between tasks in the task sequence. The error originating from this term is written as

$$\delta\epsilon_f^- = \alpha^- \sum_{d=1}^{P-1}(1-m)^{P-d} \sum_{\mu=1}^{P-d} \delta M_{\mu,\mu+d}, \tag{14}$$

where $\alpha^- \equiv -1 + (1-m)^P\left(\frac{mP}{1-m} + \frac{3-m}{2-m}\right)$ is a coefficient. $\alpha^-$ is negative if $1 > m > 0$ and $P$ is sufficiently large (Fig. 4f). Thus, to minimize the error $\delta\epsilon_f^-$, the tasks should be arrange in a way that $\delta M_{\mu,\mu+d}$ is small for small $d$, while $\delta M_{\mu,\mu+d}$ is large for large $d$. In other words, tasks following

one another in the task order sequence should be dissimilar to each other, while distant pairs should be similar.

Given a set of tasks, let us define a task dissimilarity graph by setting each task as a node and dissimilarity between two tasks as the weight of the edge between corresponding nodes (Fig. 4c). Then, a task order that learns each task only once forms a Hamiltonian path on the graph, a path that visits all nodes once but only once. We can then construct a heuristic solution for minimizing $\delta\epsilon_f^-$ by selecting a task order that yields the longest Hamiltonian path. When tasks have the same similarity with each other in terms of $C^{out}$, their similarity depends solely on $C^{in}$, allowing us to define dissimilarity as $d_{\mu\nu} \equiv 1 - C_{\mu\nu}^{in}$. Thus, the total length of the Hamiltonian path induced by a given task order follows $D_H = \sum_{\mu=1}^{P-1} d_{\mu,\mu+1}$. Consequently, $\delta\epsilon_f^-$ is rewritten as

$$\delta\epsilon_f^- = -\alpha^-\left((1-m)^{P-1}D_H + \sum_{d=2}^{P-1}(1-m)^{P-d}\sum_{\mu=1}^{P-d}d_{\mu,\mu+d}\right) + \text{const.} \tag{15}$$

Because $-\alpha^-$ is non-negative, small task dissimilarity $d_{\mu\nu}$ (i.e., large task correlation $C_{\mu\nu}^{in}$) generally helps minimizing the error. Moreover, we have $0 \leq (1-m)^{P-1} < (1-m)^{P-d}$ for $d = 2, 3, ...$, indicating $D_H$ term has the smallest impact on the error. Therefore, by choosing the largest $d_{\mu\nu}$ for $D_H$, we can make $\delta\epsilon_f^-$ small on average. We observed this trend robustly even when we sampled $\{C^{in}, C^{out}\}$ randomly (Fig. 3g). Our work thus provides theoretical insights on why the maximum Hamiltonian path provides a preferable task order, strengthening previous empirical finding (Bell & Lawrence, 2022). We call this rule as max-path rule below.

## 4.4. Application to Tasks Having Simple Graph Structures

The analyses above elucidated two principles underlying task order optimization. To illustrate these principles, we next examine task order optimization for a set of tasks with a simple task similarity structure.

Figure 4a depicts the total error estimated using Eq. 8 in a continual learning scenario involving five tasks with a chain-like similarity. We configure the input correlation matrix $C^{in}$ such that tasks $A$ and $B$ are directly correlated, while $A$ and $C$ are correlated only indirectly through $B$. Specifically, denoting the similarity between neighboring tasks on the task dissimilarity graph as $a$, we set $C_{AB}^{in} = C_{BC}^{in} = a$, $C_{AC}^{in} = a^2$, and so on (see Appendix C). Here, tasks exhibit significant overlap when $a \lesssim 1$, while tasks become independent in the limit $a \to 0$ (x-axis of Figure 4a). We set $C^{out}$ to one for all task pairs. Each line in Figure 4a

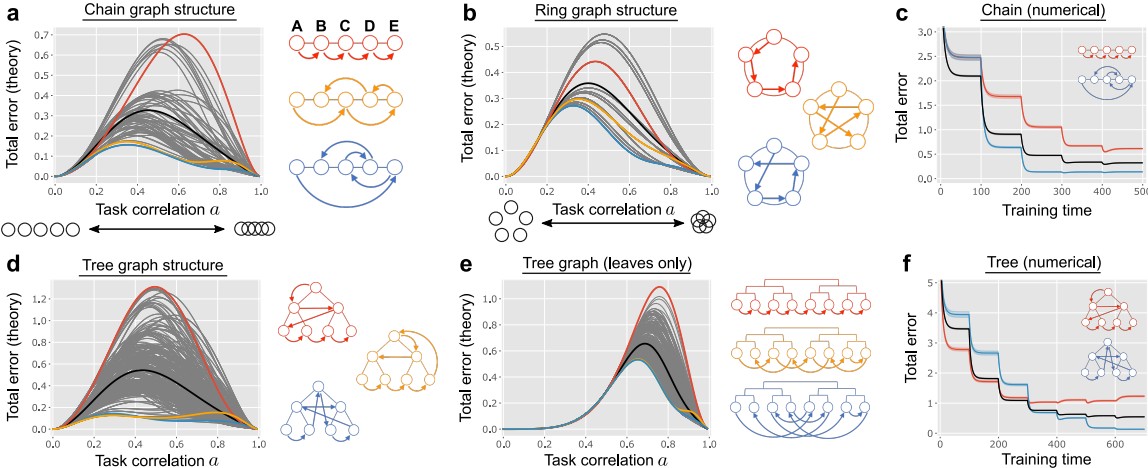

*Figure 4.* Optimal task orders for tasks with simple graph-like similarity structures. **a,b)** Total error $\epsilon_f$ under all task orders when the similarity structure of five tasks follows chain (a) and ring (b) structures. Each gray line represents one of 120 (=5!) task order, while red, orange, and blue lines highlight three representative task orders depicted on the right. Thick black line is the average error over ordering. **c)** Numerically-estimated learning dynamics of the network when tasks have a chain-graph structure. Red and blue lines represents $A{\to}B{\to}C{\to}D{\to}E$ and $A{\to}E{\to}C{\to}D{\to}B$ orders depicted in the insets. Black line is the average learning trajectory under random task ordering. **d,e)** The same as panels a and b but for tasks with similarity matrices having tree (d), and tree-leaves similarity structure (e), respectively. **f)** The same as panel c, but for tasks having tree-graph-like similarity structure.

represents the error under a specific task order. For example, the $A{\to}B{\to}C{\to}D{\to}E$ task order, depicted by the red line, consistently performed among the worst, regardless of the similarity $a$ between neighboring tasks. Surprisingly, several task orders robustly outperformed the others, such as $A{\to}C{\to}E{\to}D{\to}B$ (orange line) and $A{\to}E{\to}C{\to}D{\to}B$ (blue line). These task orders align with the two principles described earlier. First, the periphery-to-core rule suggests that the initial task should either be $A$ or $E$, as these tasks are the least typical.[1] Second, the max-path rule indicates that subsequent tasks should be as dissimilar as possible. For instance, if the first task is $A$, selecting $E$ as the second task, as in the blue line, maximizes the distance. Notably, there was approximately a seven-fold difference in performance between the best and worst task orders, underscoring the critical importance of task order in continual learning.

We observed analogous trends when applying the same analysis to tasks with ring, tree, and leaves structures (Figs. 4b, 4d, and 4e, respectively). For tasks with a tree-like similarity structure, as shown in Figure 4d, the error was minimized when tasks corresponding to leaf nodes were learned first, followed by tasks associated with root nodes (the orange and blue trees in Fig. 4d). This result aligns with the periphery-to-core rule. When only the leaf nodes were considered as tasks, as illustrated in Figure 4e, the optimal task order exhibited a complex pattern of hopping across tasks (blue tree in Fig. 4e), consistent with the max-path rule.

Numerical simulations validated the analytical results (Figs. 4c and 4f) and further revealed intricate learning dynamics. In Figure 4f, the red task order initially outperformed the black line representing the average performance, while the blue task order performed worse. However, this trend reversed around the fourth task. These findings indicate that the optimal task order is often non-trivial, and a greedy approach optimizing task-by-task error may lead to suboptimal performance.

## 5. Application to Image Classification Tasks

Our analytical investigation in the linear teacher-student setting highlighted two principles for task order optimization: the periphery-to-core rule and the max-path rule. To evaluate the potential applicability of these principles to more general settings, we next explore continual learning of image classification tasks.

### 5.1. Empirical Estimation of Task Similarity

To apply these principles, it is necessary to first measure the similarity between tasks. Here, we estimate the similarity between tasks $A$ and $B$ by measuring the zero-shot transfer performance between them (Fig. 8). Specifically, we train a network for task $A$, obtaining the learned weights $W_A$. We then measure the error of this trained network on task $B$, denoted as $\epsilon_B[W_A]$. Since the transfer performance from task $A$ to $B$ generally differs from that of $B$ to $A$ we take

---

[1]Due to mirror symmetry, the $E{\to}A{\to}C{\to}B{\to}D$ order exhibits equivalent performance to $A{\to}E{\to}C{\to}D{\to}B$.

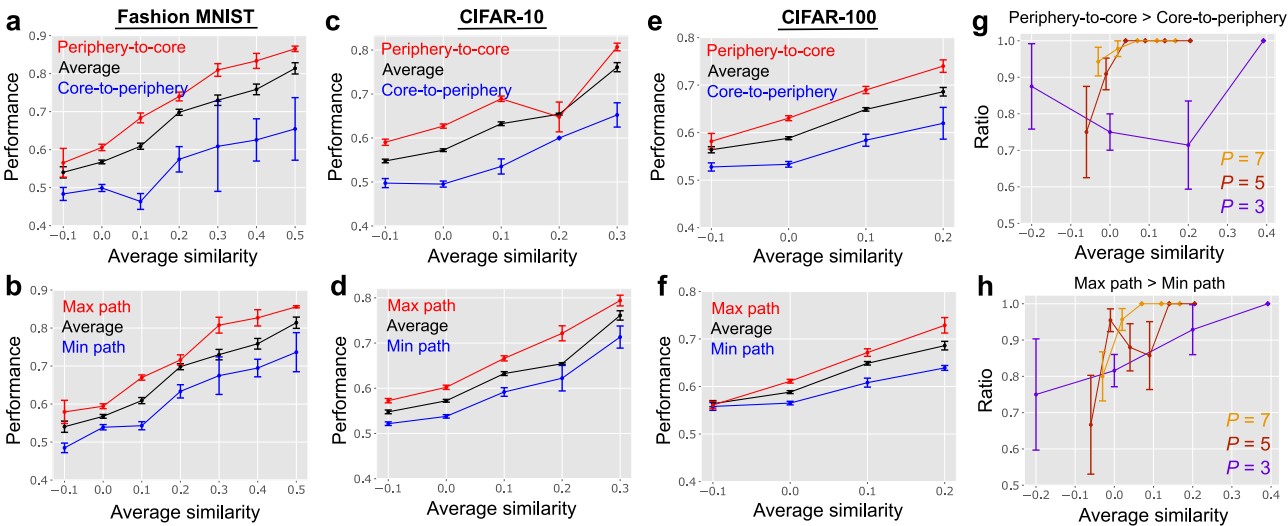

*Figure 5.* Task order preference in continuous image classification tasks. **a–f)** Continual learning performance, defined as the average test accuracy across all the tasks after learning, under various task orders. Panels (a, c, e) compare the periphery-to-core rule against the core-to-periphery rule, whereas panels (b, d, f) compare the max-path rule with the min-path rule. **g, h)** The ratio of task sets where the periphery-to-core rule outperforms the core-to-periphery rule (g), and where the max-path rule outperforms the min-path rule (h), under CIFAR-100. Different colors represent results for different numbers of tasks ($P = 3, 5, 7$). See Appendix C.3 for details.

the mean of both directions and define the similarity $\rho_{AB}$:

$$\rho_{AB} = 1 - \frac{1}{2}\left(\sqrt{\frac{\epsilon_B[W_A]}{\epsilon_{B,sf}[W_A]}} + \sqrt{\frac{\epsilon_A[W_B]}{\epsilon_{A,sf}[W_B]}}\right). \quad (16)$$

Here, $\epsilon_{B,sf}[W_A]$ represents the error on task $B$ with label shuffling, which serves as the chance-level error. The square root is taken because the error scales with the squared value of task correlation in our linear model (see Appendix C.4).

Although this method requires training the network on all $P$ tasks, the computational complexity of training is $\mathcal{O}(P)$, which is significantly smaller than the naive task order optimization that requires a computational cost of $\mathcal{O}(P!)$. Furthermore, this method only requires the inputs and outputs of the trained network, making it applicable even in situations where the model's internal details are inaccessible.

### 5.2. Numerical Results

We estimated the performance of the periphery-to-core rule and max-path rule in task-incremental continual learning using Fashion-MNIST (Xiao et al., 2017), CIFAR-10, and CIFAR-100 dataset (Krizhevsky et al., 2009) (see Appendix C for the details). For the Fashion-MNIST and CIFAR-10 datasets, we randomly generated five binary image classification tasks by dividing 10 labels into 5 pairs without replacement. In the case of CIFAR-100, we selected 10 labels out of 100 labels randomly and generated 5 binary classifications. For Fashion-MNIST, we trained a multi-layered perceptron with two hidden layers, while for CIFAR-10/100, we used a convolutional neural network with two convolu-

tional layers and one dense layer, to explore robustness against the model architecture.

We found that the final performance was modulated by the estimated average similarity among tasks, $\bar{\rho} = \frac{1}{P(P-1)}\sum_{\mu \neq \nu} \rho_{\mu\nu}$, we thus plotted the performance of each task-order rule as a function of the average similarity (Fig. 5). In all three settings, we found that the periphery-to-core rule robustly outperforms the core-to-periphery rule and average performance over random ordering (Fig. 5a,c,e). Similarly, the max-path rule outperformed both the min-path rule and the random ordering (Fig. 5b,d,e; see also Bell & Lawrence (2022)). Moreover, we observed consistent results under a continual learning of a multi-class classification (Fig.7a and b). The periphery-to-core rule outperformed the max-path rule on average, but the difference was small (red lines in Fig. 5 top vs bottom). When we increased the number of binary classification tasks from 3 to 7 using CIFAR-100, the performance advantage periphery-to-core over core-to-periphery increased (Fig. 5g) as expected from the linear model (Fig. 6c). This was not evident for max-path and min-path rules potentially because the difference was already high under $P = 3$ (Fig. 5h).

We also investigated the inference of task similarity and ordering from a small subset of training data. This extension is crucial, as it demonstrates the practical relevance of our theory to real-world machine learning settings where full access to all training data upfront is often unrealistic (in contrast, when complete data is available, naive multi-task learning may suffice). When we reduced the number of training samples used to estimate task similarity across tasks

in CIFAR-10, the relative advantage of both the periphery-to-core rule and the max-path rule over a random task order remained robust. Even when only 1% of the training data was used for estimating task similarity, we observed a performance gain comparable to that in the full data scenario (Panels c and d vs. g and h in Fig. 9). However, when the amount of data was reduced to 0.1% (approximately 10 samples per task), the performance gain became non-significant. We observed similar trends with both the Fashion-MNIST and CIFAR-100 datasets, although the results for CIFAR-100 were less robust, particularly in the negative similarity regime (Fig. 10). These results suggest the robustness of the task order optimization principles found in our simple analysis.

## 6. Discussion

In this work, we derived a simple analytical expression to explain how task similarity and ordering influence continual learning performance in a linear model with latent structure. Based on this result, we proposed two principles for task order optimization: the periphery-to-core rule and the max-path rule, the latter of which was predicted by Bell & Lawrence (2022). We validated these principles in task-incremental continual image classification tasks using both multi-layer perceptrons and convolutional neural networks. Thus, this work proposes basic principles for task order optimization in the context of continual learning for multiple tasks.

### Limitations

Our theoretical results were derived in a linear model under the assumption of random task generation, which limits their direct applicability. However, we numerically confirmed that the proposed ordering rules hold in both convolutional neural networks and multi-layer perceptrons trained for continual image recognition tasks. Future work should further evaluate these rules in domains closer to real-world applications, including deep-RL, robotics, and language models. Additionally, in this work, we restricted the model setting to scenarios where each task is learned only once and trained to convergence. The first assumption can be readily relaxed as long as the total number of tasks remains small (see Appendix A.4). Relaxing the second assumption is an important direction for future work.

### Acknowledgements

This work was partially supported by McDonnell Center for Systems Neuroscience.

## Impact Statement

Task order optimization for continual learning of multiple tasks may potentially contribute beyond the field of machine learning from school curriculum design (Rafferty et al., 2016; Zhu et al., 2018) to animal training protocol in neuroscience experiments (Krueger & Dayan, 2009). Nevertheless, due to theoretical nature of this work, there are no specific societal consequence that we feel must be highlighted here.

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

# A. Analysis of the Impact of Task Similarity on Continual Learning

## A.1. Model Setting

Below, we analyze task-order dependence of continual learning using linear teacher-student models with a latent factor. In teacher-student models, the generative model of the task parameterized explicitly by the teacher model, making the learning dynamics and the performance analytically tractable (Gardner & Derrida, 1989; Saad & Solla, 1995; Zdeborová & Krzakala, 2016). Here, the generative model for input $\boldsymbol{x} \in \mathbb{R}^{N_x}$ and the target output $\boldsymbol{y} \in \mathbb{R}^{N_y}$ is constructed as

$$\boldsymbol{s} \sim \mathcal{N}(0, I), \quad \boldsymbol{x} = A_\mu \boldsymbol{s}, \quad \boldsymbol{y}^* = B_\mu \boldsymbol{s}, \tag{17}$$

where $\boldsymbol{s} \in \mathbb{R}^{N_s}$ is the latent variable that underlies $\boldsymbol{x}$ and $\boldsymbol{y}^*$, $I$ is the identity matrix, and $A_\mu \in \mathbb{R}^{N_x \times N_s}$ and $B_\mu \in \mathbb{R}^{N_y \times N_s}$ are mixing matrices for the input and the target output at task $\mu = 1, ..., P$, respectively.

We generate matrices $\{A_\mu, B_\mu\}_{\mu=1}^P$ randomly but with task-to-task correlation. We specify the element-wise correlation among input generation matrices $\{A_\mu\}_{\mu=1}^P$ by a $P \times P$ correlation matrix $C^{in}$ and specify the correlation among the target output generation matrices $\{B_\mu\}_{\mu=1}^P$ by another $P \times P$ correlation matrix $C^{out}$. $C^{in}$ and $C^{out}$ are constrained to be correlation matrices, but arbitrary otherwise. For all $1 \leq i \leq N_x$ and $1 \leq j \leq N_s$, we generate $(i, j)$-th elements of matrices $A_1, ..., A_P$ by jointly sampling them from a Gaussian distribution with mean zero and covariance $\frac{1}{N_s} C^{in}$:

$$\begin{pmatrix} A_{ij}^1 \\ \vdots \\ A_{ij}^P \end{pmatrix} \sim \mathcal{N}\left(\boldsymbol{0}, \tfrac{1}{N_s} C^{in}\right). \tag{18}$$

Similarly, we generate $(i, j)$-th elements of matrices $B_1, ..., B_P$ by $(B_{ij}^1, ..., B_{ij}^P)^T \sim \mathcal{N}\left(\boldsymbol{0}, \tfrac{1}{N_s} C^{out}\right)$. Note that, under this construction, two different elements in a matrix $A_\mu$ are independent with each other, but the same element in matrices for two different tasks are correlated with each other. Although the random task generation assumption limits direct applicability of our theory, it enables us to obtain insights into how task similarity influences the overall performance and optimal task order. Moreover, our analytical results up to Eq. 28 hold for arbitrary mixing matrices $\{A_\mu, B_\mu\}$, as they don't assume the expectation over $\{A_\mu, B_\mu\}$.

The student network that learns the task is specified to be a linear network:

$$\boldsymbol{y} = W\boldsymbol{x}, \tag{19}$$

where $W \in \mathbb{R}^{N_y \times N_x}$ is the trainable weight. The mean-squared error between the output of the student network $\boldsymbol{y}$ and the target for the $\mu$-th task $\boldsymbol{y}^*$ is given by

$$\epsilon_\mu[W] \equiv \tfrac{1}{N_y} \left\langle \|\boldsymbol{y} - \boldsymbol{y}^*\|^2 \right\rangle_{\boldsymbol{s}} = \tfrac{1}{N_y} \|B_\mu - W A_\mu\|_F^2 \tag{20}$$

The second equality follows from the Gaussianity of $\boldsymbol{s}$. We consider task-incremental continual learning (Van de Ven & Tolias, 2019) where the network is trained for task $\mu = 1, ..., P$ in sequence. During the training for the $\mu$-th task, weight $W$ is updated by gradient descent on error $\epsilon_\mu$:

$$W \leftarrow W - \eta \frac{\partial \epsilon_\mu[W]}{\partial W} = W - \frac{2\eta}{N_y}(B_\mu - W A_\mu)A_\mu^T. \tag{21}$$

We denote the weight after training on task $\mu$ as $W_\mu$. The total error on all tasks at the end of all $P$ task learning becomes:

$$\epsilon_f \equiv \sum_{\mu=1}^P \epsilon_\mu[W_P] = \frac{1}{N_y} \sum_{\mu=1}^P \|B_\mu - W_P A_\mu\|_F^2. \tag{22}$$

To uncover how similarity between tasks influences the final error, in this work, we focus on the average performance over randomly generated mixing matrices $\{A_\mu, B_\mu\}_{\mu=1}^P$ under a fixed pair of task correlation matrices $C^{in}, C^{out}$. We define the average of the final error $\epsilon_f$ over generative models $\{A_\mu, B_\mu\}_{\mu=1}^P$ by

$$\bar{\epsilon}_f \equiv \langle \epsilon_f \rangle_{A,B}. \tag{23}$$

Below, we first derive the analytical expression of $W_\mu$ then estimate the total final error $\bar{\epsilon}_f$.

## A.2. The Weight after Continual Learning of $P$ Tasks

Considering the gradient flow limit of learning dynamics, the weight update (Eq. 21) is rewritten as

$$\frac{dW}{dt} = -(B_\mu - WA_\mu)A_\mu^T. \tag{24}$$

Let us denote singular value decomposition (SVD) of $A_\mu$ by $A_\mu = U_\mu \Lambda_\mu V_\mu^T$, where $U_\mu \in \mathbb{R}^{N_x \times N_o}$ and $V_\mu \in \mathbb{R}^{N_s \times N_o}$ are semi-orthonormal matrices (i.e., $U^T U = V^T V = I$) and $\Lambda_\mu \in \mathbb{R}^{N_o \times N_o}$ is a non-negative diagonal matrix. Then, at any point during learning, there exists a matrix $Q(t) \in \mathbb{R}^{N_y \times N_o}$ such that $W(t)$ is written as $W(t) = W_{\mu-1} + Q(t)U_\mu^T$ because weight change during learning of the $\mu$-th task is constrained to the space spanned by $U_\mu^T$. Thus, at the convergence of learning, $\frac{dW}{dt} = 0$, we have

$$\left(B_\mu - [W_{\mu-1} + Q_\mu U_\mu^T]A_\mu\right)A_\mu^T = 0. \tag{25}$$

Solving this equation with respect to $Q_\mu$, we get $Q_\mu = B_\mu V_\mu \Lambda_\mu^{-1} - W_{\mu-1}U_\mu$. Therefore, the weight after training on the $\mu$-th task becomes

$$W_\mu = W_{\mu-1}(I - U_\mu U_\mu^T) + B_\mu A_\mu^+, \tag{26}$$

where $A_\mu^+$ is the pseudo-inverse of $A_\mu$ ($A_\mu^+ = V_\mu \Lambda_\mu^{-1} U_\mu^T$). By applying this result iteratively from zero initialization, $W_\mu$ is rewritten as

$$W_\mu = \sum_{\nu=1}^{\mu}(B_\nu A_\nu^+) \prod_{\rho=\nu+1}^{\mu} (I - U_\rho U_\rho^T), \tag{27}$$

where $\prod_{\rho=\nu+1}^{\mu}(I - U_\rho U_\rho^T)$ is the identity matrix if $\mu = \nu$, otherwise,

$$\prod_{\rho=\nu+1}^{\mu} (I - U_\rho U_\rho^T) = (I - U_{\nu+1}U_{\nu+1}^T)(I - U_{\nu+2}U_{\nu+2}^T) \cdots (I - U_\mu U_\mu^T). \tag{28}$$

To further investigate how task similarity impacts continual learning performance, below we focus on the large $N_x$ regime, and analyze the learning behavior at $\frac{N_s}{N_x} \to 0$ limit. This assumption of the presence of low-dimensional latent factor is consistent with many real-world datasets (Yu et al., 2017; Cohen et al., 2020). If $A_\mu$ is a very-tall random matrix (i.e., $N_x \gg N_s$), pseudo-inverse $A_\mu^+$ is approximated by a scaled transpose $\gamma A_\mu^T$, and $U_\mu U_\mu^T$ approximately follows $U_\mu U_\mu^T \approx \gamma A_\mu A_\mu^T$, where $\gamma = \frac{N_s}{N_x}$ (see Appendix A.4). Thus, we have

$$W_\mu \approx \gamma \sum_{\nu=1}^{\mu}(B_\nu A_\nu^T) \prod_{\rho=\nu+1}^{\mu} (I - \gamma A_\rho A_\rho^T). \tag{29}$$

Using the approximation from Eq. 29, the error on the $\mu$-th task after training on $\nu$-th task, $\epsilon_\mu[W_\nu]$, is

$$\epsilon_\mu[W_\nu] = \|B_\mu - W_\nu A_\mu\|_F^2 \approx \left\|B_\mu - \gamma \sum_{\rho=1}^{\nu}(B_\rho A_\rho^T) \prod_{\sigma=\rho+1}^{\nu} (I - \gamma A_\sigma A_\sigma^T)A_\mu\right\|_F^2. \tag{30}$$

## A.3. Proof of Theorem 3.1

Substituting $W_P$ with Eq. 29, at $\frac{N_s}{N_x} \to 0$ limit, $\bar{\epsilon}_f$ is rewritten as

$$
\begin{aligned}
\bar{\epsilon}_f &= \frac{1}{N_y}\sum_{\mu=1}^{P}\left\langle \left\| B_\mu - \gamma \sum_{\rho=1}^{P} B_\rho A_\rho^T \prod_{\sigma=\rho+1}^{P}(I - \gamma A_\sigma A_\sigma^T)A_\mu \right\|_F^2 \right\rangle \\
&= \frac{1}{N_y}\sum_{\mu=1}^{P}\left\langle \|B_\mu\|_F^2 \right\rangle - \frac{2\gamma}{N_y}\sum_{\mu=1}^{P}\sum_{\rho=1}^{P}\left\langle \text{tr}\left[ B_\mu^T B_\rho A_\rho^T \prod_{\sigma=\rho+1}^{P}(I - \gamma A_\sigma A_\sigma^T)A_\mu \right] \right\rangle \\
&\quad + \frac{\gamma^2}{N_y}\sum_{\mu=1}^{P}\left\langle \left\| \sum_{\rho=1}^{P} B_\rho A_\rho^T \prod_{\sigma=\rho+1}^{P}(I - \gamma A_\sigma A_\sigma^T)A_\mu \right\|_F^2 \right\rangle
\end{aligned}
\tag{31}
$$

Taking expectation over $\{A_\mu, B_\mu\}_{\mu=1}^P$, the first term is $\left\langle \|B_\mu\|_F^2 \right\rangle = N_y$. The second term is rewritten as

$$
\frac{\gamma}{N_y} \sum_{\mu=1}^P \sum_{\rho=1}^P \left\langle \mathrm{tr}\left[ B_\mu^T B_\rho A_\rho^T \prod_{\sigma=\rho+1}^P (I - \gamma A_\sigma A_\sigma^T) A_\mu \right] \right\rangle
$$

$$
= \frac{1}{N_x} \sum_{\mu=1}^P \sum_{\rho=1}^P C_{\mu\rho}^{out} \left\langle \mathrm{tr}\left[ A_\rho^T (I - \gamma A_{\rho+1} A_{\rho+1}^T) \cdots (I - \gamma A_P A_P^T) A_\mu \right] \right\rangle
$$

$$
= \frac{1}{N_x} \sum_{\mu=1}^P \sum_{\rho=1}^P C_{\mu\rho}^{out} \left\langle \mathrm{tr}[A_\rho^T A_\mu] - \gamma \sum_{\sigma_1=\rho+1}^P \mathrm{tr}[A_\rho^T A_{\sigma_1} A_{\sigma_1}^T A_\mu] + \gamma^2 \sum_{\sigma_1=\rho+1}^{P-1} \sum_{\sigma_2=\sigma_1+1}^P \mathrm{tr}[A_\rho^T A_{\sigma_1} A_{\sigma_1}^T A_{\sigma_2} A_{\sigma_2}^T A_\mu] - ... \right\rangle
$$

$$
= \frac{1}{N_x} \sum_{\mu=1}^P \sum_{\rho=1}^P C_{\mu\rho}^{out} \sum_{k=0}^{P-\rho} (-\gamma)^k \sum_{\rho<\sigma_1<...<\sigma_k\leq P} \left\langle \mathrm{tr}\left[ A_\rho^T A_{\sigma_1} A_{\sigma_1}^T ... A_{\sigma_k} A_{\sigma_k}^T A_\mu \right] \right\rangle. \tag{32}
$$

In the first line, we took expectation over $\{B_\mu\}$, which yields $\frac{\gamma}{N_y} \left\langle \mathrm{tr}[B_\mu^T B_\rho M] \right\rangle_B = \frac{N_s/N_x}{N_y} \frac{N_y C_{\mu\rho}^B}{N_s} \mathrm{tr}[M] = \frac{C_{\mu\rho}^B}{N_x} \mathrm{tr}[M]$ for arbitrary matrix $M$. In the third line, we rearranged the terms inside the trace based on $\gamma$ dependence. The summation $\sum_{\rho<\sigma_1<...<\sigma_k\leq P}$ in the last line is summation over a set of indices $\sigma_1, \sigma_2, ..., \sigma_k$ that satisfy $\rho < \sigma_1 < \sigma_2 < ... < \sigma_k \leq P$ condition. Under $N_x \gg N_s$, the expectation term in the equation above follows (see Appendix A.4)

$$
\left\langle \gamma^k \mathrm{tr}\left[ A_\rho^T A_{\sigma_1} A_{\sigma_1}^T ... A_{\sigma_k} A_{\sigma_k}^T A_\mu \right] \right\rangle = N_x \left( C_{\rho\sigma_1}^{in} C_{\sigma_1\sigma_2}^{in} ... C_{\sigma_k\mu}^{in} + \mathcal{O}\left( \frac{N_s}{N_x} \right) \right). \tag{33}
$$

Moreover, if we define an upper-triangle matrix $C^{in,U} \in \mathbb{R}^{P \times P}$ by

$$
C^{in,U} = \begin{cases} C_{\mu\nu}^{in} & (\text{if} \quad \mu < \nu) \\ 0 & (\text{otherwise}) \end{cases}, \tag{34}
$$

we have

$$
\sum_{\rho<\sigma_1<...<\sigma_k\leq P} \left\langle \mathrm{tr}\left[ A_\rho^T A_{\sigma_1} A_{\sigma_1}^T ... A_{\sigma_k} A_{\sigma_k}^T A_\mu \right] \right\rangle = \sum_{\rho<\sigma_1<...<\sigma_k\leq P} C_{\rho\sigma_1}^{in} C_{\sigma_1\sigma_2}^{in} ... C_{\sigma_k\mu}^{in} + \mathcal{O}\left( \frac{N_s}{N_x} \right)
$$

$$
= \left[ \left( C^{in,U} \right)^k C^{in} \right]_{\rho\mu} + \mathcal{O}\left( \frac{N_s}{N_x} \right) \tag{35}
$$

The last line follows because the upper triangle matrix $C^{in,U}$ satisfies

$$
\sum_{\sigma_1=1}^P \sum_{\sigma_2=1}^P ... \sum_{\sigma_k=1}^P C_{\rho\sigma_1}^{in,U} C_{\sigma_1\sigma_2}^{in,U} ... C_{\sigma_{k-1}\sigma_k}^{in,U} C_{\sigma_k\mu}^{in} = \sum_{\sigma_1=\rho+1}^P \sum_{\sigma_2=\sigma_1+1}^P ... \sum_{\sigma_k=\sigma_{k-1}+1}^P C_{\rho\sigma_1}^{in,U} C_{\sigma_1\sigma_2}^{in,U} ... C_{\sigma_{k-1}\sigma_k}^{in,U} C_{\sigma_k\mu}^{in}
$$

$$
= \sum_{\rho<\sigma_1<...<\sigma_k\leq P} C_{\rho\sigma_1}^{in} C_{\sigma_1\sigma_2}^{in} ... C_{\sigma_{k-1}\sigma_k}^{in} C_{\sigma_k\mu}^{in}. \tag{36}
$$

Moreover, because $\left[ (C^{in,U})^k \right]_{\rho\nu} = 0$ for any $\nu$ if $\rho$ satisfies $\rho \geq P - k + 1$, we have

$$
\sum_{k=0}^{P-\rho} \left[ \left( C^{in,U} \right)^k C^{in} \right]_{\rho\mu} = \sum_{k=0}^P \left[ \left( C^{in,U} \right)^k C^{in} \right]_{\rho\mu} \tag{37}
$$

Therefore, taking $\frac{N_s}{N_x} \to 0$ limit, it follows that

$$\frac{1}{N_x} \sum_{\mu=1}^{P} \sum_{\rho=1}^{P} C_{\mu\rho}^{out} \sum_{k=0}^{\nu-\rho} (-\gamma)^k \sum_{\rho<\sigma_1<...<\sigma_k\leq P} \left\langle \text{tr} \left[ A_\rho^T A_{\sigma_1} A_{\sigma_1}^T ... A_{\sigma_k} A_{\sigma_k}^T A_\mu \right] \right\rangle$$

$$\approx \sum_{\mu=1}^{P} \sum_{\rho=1}^{P} C_{\mu\rho}^{out} \sum_{k=0}^{P} (-1)^k \left[ \left( C^{in,U} \right)^k C^{in} \right]_{\rho\mu}$$

$$= \sum_{\mu=1}^{P} \sum_{\rho=1}^{P} C_{\mu\rho}^{out} \left[ \left( I + C^{in,U} \right)^{-1} C^{in} \right]_{\rho\mu}$$

$$= \text{tr} \left[ C^{out} \left( I + C^{in,U} \right)^{-1} C^{in} \right]. \tag{38}$$

In the third line, we used

$$(I + C^{in,U}) \sum_{k=0}^{P} (-1)^k \left( C^{in,U} \right)^k = I + (-1)^P \left( C^{in,U} \right)^{P+1} = I. \tag{39}$$

We can evaluate the third term of Eq. 31 in an analogous manner:

$$\frac{\gamma^2}{N_y} \sum_{\mu=1}^{P} \left\langle \left\| \sum_{\rho=1}^{P} B_\rho A_\rho^T \prod_{\sigma=\rho+1}^{P} (I - \gamma A_\sigma A_\sigma^T) A_\mu \right\|_F^2 \right\rangle$$

$$= \frac{\gamma^2}{N_y} \sum_{\mu=1}^{P} \sum_{\rho=1}^{P} \sum_{\rho'=1}^{P} \left\langle \text{tr} \left[ B_{\rho'}^T B_\rho A_\rho^T \prod_{\sigma=\rho+1}^{P} (I - \gamma A_\sigma A_\sigma^T) A_\mu \left( A_{\rho'}^T \prod_{\sigma'=\rho'+1}^{P} (I - \gamma A_{\sigma'} A_{\sigma'}^T) A_\mu \right)^T \right] \right\rangle$$

$$= \frac{N_s}{N_x^2} \sum_{\mu=1}^{P} \sum_{\rho=1}^{P} \sum_{\rho'=1}^{P} C_{\rho\rho'}^{out} \left\langle \text{tr} \left[ A_\rho^T (I - \gamma A_{\rho+1} A_{\rho+1}^T)...(I - \gamma A_P A_P^T) A_\mu A_\mu^T (I - \gamma A_P A_P^T)...(I - \gamma A_{\rho'+1} A_{\rho'+1}^T) A_{\rho'} \right] \right\rangle. \tag{40}$$

The term inside the trace can be expanded as

$$\left\langle \text{tr} \left[ A_\rho^T (I - \gamma A_{\rho+1} A_{\rho+1}^T)...(I - \gamma A_P A_P^T) A_\mu A_\mu^T (I - \gamma A_P A_P^T)...(I - \gamma A_{\rho'+1} A_{\rho'+1}^T) A_{\rho'} \right] \right\rangle$$

$$= \sum_{k=1}^{P-\rho} \sum_{k'=1}^{P-\rho'} \sum_{\rho<\sigma_1<...<\sigma_k\leq P} \sum_{\rho'<\sigma'_1<...<\sigma'_k\leq P} (-\gamma)^{k+k'} \left\langle \text{tr} \left[ A_\rho^T A_{\sigma_1} A_{\sigma_1}^T ... A_{\sigma_k} A_{\sigma_k}^T A_\mu A_\mu^T A_{\sigma'_k} A_{\sigma'_k}^T ... A_{\sigma'_1} A_{\sigma'_1}^T A_{\rho'} \right] \right\rangle, \tag{41}$$

and the expectation over $\{A_\mu\}$ follows (Appendix A. 4)

$$\gamma^{k+k'} \left\langle \text{tr} \left[ A_\rho^T A_{\sigma_1} A_{\sigma_1}^T ... A_{\sigma_k} A_{\sigma_k}^T A_\mu A_\mu^T A_{\sigma'_k} A_{\sigma'_k}^T ... A_{\sigma'_1} A_{\sigma'_1}^T A_{\rho'} \right] \right\rangle = \frac{N_x^2}{N_s} \left( C_{\rho\sigma_1}^{in} C_{\sigma_1\sigma_2}^{in} ... C_{\sigma_k\mu}^{in} C_{\mu\sigma'_k}^{in} ... C_{\sigma'_2\sigma'_1}^{in} C_{\sigma'_1\rho'}^{in} + \mathcal{O} \left( \frac{N_s}{N_x} \right) \right). \tag{42}$$

Therefore, at $\frac{N_s}{N_x} \to 0$ limit, the squared term is evaluated as

$$\frac{\gamma^2}{N_y} \sum_{\mu=1}^{P} \left\langle \left\| \sum_{\rho=1}^{P} B_\rho A_\rho^T \prod_{\sigma=\rho+1}^{P} (I - \gamma A_\sigma A_\sigma^T) A_\mu \right\|_F^2 \right\rangle$$

$$= \sum_{\mu=1}^{P} \sum_{\rho=1}^{P} \sum_{\rho'=1}^{P} C_{\rho\rho'}^{out} \left( \sum_{k=0}^{P-\rho} (-1)^k \sum_{\rho<\sigma_1<...<\sigma_k\leq P} C_{\rho\sigma_1}^{in}...C_{\sigma_k\mu}^{in} \right) \left( \sum_{k'=0}^{P-\rho'} (-1)^{k'} \sum_{\rho'<\sigma'_1<...<\sigma'_{k'}\leq P} C_{\rho'\sigma'_1}^{in}...C_{\sigma'_{k'}\mu}^{in} \right)$$

$$= \sum_{\mu=1}^{P} \sum_{\rho=1}^{P} \sum_{\rho'=1}^{P} C_{\rho\rho'}^{out} \left[ (I + C^{in,U})^{-1} C^{in} \right]_{\rho\mu} \left[ (I + C^{in,U})^{-1} C^{in} \right]_{\rho'\mu}$$

$$= \text{tr} \left[ C^{out} (I + C^{in,U})^{-1} C^{in} \left( (I + C^{in,U})^{-1} C^{in} \right)^T \right]. \tag{43}$$

Noticing that the first term of Eq. 31 is rewritten as

$$\frac{1}{N_y} \sum_{\mu=1}^{P} \left\langle \|B_\mu\|_F^2 \right\rangle = P = \text{tr}[C^{out}], \tag{44}$$

at $\frac{N_s}{N_x} \to 0$ limit, the final error $\bar{\epsilon}_f$ is written as

$$\bar{\epsilon}_f = \text{tr}[C^{out}] - 2\text{tr}[C^{out}(I + C^{in,U})^{-1}C^{in}] + \text{tr}\left[C^{out}(I + C^{in,U})^{-1}C^{in}\left((I + C^{in,U})^{-1}C^{in}\right)^T\right]$$
$$= \left\|(C^{out})^{1/2}\left(I - (I + C^{in,U})^{-1}C^{in}\right)\right\|_F^2. \tag{45}$$

Thus, we obtained the equality in Theorem 3.1. Note that because $C^{out}$ is a correlation matrix, there exists a matrix $(C^{out})^{1/2}$ such that $(C^{out})^{1/2}(C^{out})^{1/2} = C^{out}$. Because $C^{in} = I + C^{in,U} + (C^{in,U})^T$, $\bar{\epsilon}_f$ is also written as

$$\bar{\epsilon}_f = \left\|(C^{out})^{1/2}(I + C^{in,U})^{-1}\left(C^{in,U}\right)^T\right\|_F^2. \tag{46}$$

Note that, if $C^{in} = I$, the error is zero. This is consistent with previous results showing that in the absence of overlap between tasks, continual learning doesn't suffer from forgetting (Ramasesh et al.; Lee et al., 2021; Peng et al., 2023). Additionally, Eq. 33 requires $P \ll \frac{N_x}{N_s}$ (see Appendix A).4 below), thus the obtained expression doesn't hold when the number of tasks is comparable to the network size.

### A.4. Expectation over Random Correlated Matrices $\{A_\mu\}$

We first show that $A_\mu^+ \to \gamma A_\mu^T$ and $U_\mu U_\mu^T \to \gamma A_\mu A_\mu^T$ at $\frac{N_s}{N_x} \to 0$, where $\gamma = \frac{N_s}{N_x}$ and $U_\mu$ is defined by SVD of $A_\mu$, $A_\mu = U_\mu \Lambda_\mu V_\mu^T$. If $\Lambda_\mu = \frac{1}{\sqrt{\gamma}}I$, then we have $A_\mu A_\mu^T = U_\mu \Lambda_\mu^2 U_\mu^T = \frac{1}{\gamma} U_\mu U_\mu^T$, and

$$A_\mu^+ = V_\mu \Lambda_\mu^{-1} U_\mu^T = \sqrt{\gamma}\left(U_\mu V_\mu^T\right)^T = \gamma\left(\frac{1}{\sqrt{\gamma}} U_\mu V_\mu^T\right)^T = A_\mu^T. \tag{47}$$

Thus, it is sufficient to show that $\Lambda_\mu \to \frac{1}{\sqrt{\lambda}}I$ at $\frac{N_s}{N_x} \to 0$. The mean and variance of $N_s \times N_s$ matrix $A_\mu^T A_\mu$ over randomly sampled $A_\mu$ obey

$$\left\langle A_\mu^T A_\mu \right\rangle_A = \frac{N_x}{N_s} I$$
$$\left\langle [A_\mu^T A_\mu - \frac{1}{\gamma}I] \odot [A_\mu^T A_\mu - \frac{1}{\gamma}I] \right\rangle = \frac{N_x}{N_s^2}\left(I + \mathbf{1}\mathbf{1}^T\right), \tag{48}$$

where $\mathbf{1}$ is a all-one vector and $\odot$ represents Hadamard product, indicating that the standard deviation of $A_\mu^T A_\mu$ shows $\mathcal{O}\left(\frac{N_s}{N_x}\right)$ scaling with respect to the mean. Thus, $\gamma A_\mu^T A_\mu \to I$ at $\frac{N_s}{N_x} \to 0$, implying $\Lambda_\mu \to \frac{1}{\sqrt{\lambda}}I$ at $\frac{N_s}{N_x} \to 0$.

Regarding expectation over $A$ in Eq. 33, expanding the equation up to the next to the leading order, we have

$$\left\langle \text{tr}\left[A_\rho^T A_{\sigma_1} A_{\sigma_1}^T ... A_{\sigma_k} A_{\sigma_k}^T A_\mu\right] \right\rangle$$
$$= \sum_{i_0,...,i_k} \sum_{j_0,...,j_k} \left\langle A_{i_0 j_0}^\rho A_{i_0 j_1}^{\sigma_1} A_{i_1 j_1}^{\sigma_1} A_{i_1 j_2}^{\sigma_2} A_{i_2 j_2}^{\sigma_1} ... A_{i_{k-1} j_k}^{\sigma_k} A_{i_k j_k}^{\sigma_k} A_{i_k j_0}^\mu \right\rangle$$
$$= \sum_{i_0,...,i_k} \sum_{j_0,...,j_k} \left(\delta_{j_0,j_1,j_2,...,j_k}\left(\frac{1}{N_s}\right)^{k+1} C_{\rho\sigma_1}^{in} C_{\sigma_1\sigma_2}^{in} ... C_{\sigma_k\mu}^{in}\right.$$
$$\left. + \sum_l \delta_{i_{l-1}i_l}\delta_{j_0,j_1,...,j_{l-1},j_{l+1},...,j_k}\left(\frac{1}{N_s}\right)^{k+1} C_{\rho\sigma_1}^{in} C_{\sigma_1\sigma_2}^{in} ... C_{\sigma_{l-2}\sigma_{l-1}}^{in} C_{\sigma_{l-1}\sigma_{l+1}}^{in} C_{\sigma_l\sigma_l}^{in} C_{\sigma_{l+1}\sigma_{l+2}}^{in} ... C_{\sigma_k\mu}^{in} + ...\right)$$
$$= N_x\left(\frac{N_x}{N_s}\right)^k C_{\rho\sigma_1}^{in} C_{\sigma_1\sigma_2}^{in} ... C_{\sigma_k\mu}^{in} + N_s\left(\frac{N_x}{N_s}\right)^k \sum_l C_{\rho\sigma_1}^{in} C_{\sigma_1\sigma_2}^{in} ... C_{\sigma_{l-1}\sigma_{l+1}}^{in} C_{\sigma_l\sigma_l}^{in} ... C_{\sigma_k\mu}^{in} + \mathcal{O}\left(\left(\frac{N_x}{N_s}\right)^k\right)$$
$$= \gamma^{-k} N_x\left(C_{\rho\sigma_1}^{in} C_{\sigma_1\sigma_2}^{in} ... C_{\sigma_k\mu}^{in} + \mathcal{O}\left(\frac{N_s}{N_x}\right)\right) \tag{49}$$

$\delta_{j_0, j_1, \ldots, j_k}$ in the third line is the Kronecker delta function that returns 1 if $j_0 = j_1 = \ldots = j_k$, otherwise returns 0. The third line follows from Isserlis' theorem, which states that the expectation over multivariate normal variables can be decomposed into summation over all pair-wise partitions (Helias & Dahmen, 2020). In the equation above, the partition that pairs neighboring matrices takes $\mathcal{O}(N_x^{k+1})$ value, while all other partitions yield $\mathcal{O}(N_x^k)$ value at most because of indices mismatch. Note that, the number of second order terms depends on $P$, as suggested by the summation over $l$ in the third line. Thus, we expect that our theory hold only when $P$ satisfies $P \ll \frac{N_x}{N_s}$.

From a parallel argument with the one above, the expectation in Eq. 42 is evaluated as

$$
\begin{aligned}
\gamma^{k+k'} & \left\langle \text{tr} \left[ A_\rho^T A_{\sigma_1} A_{\sigma_1}^T \ldots A_{\sigma_k} A_{\sigma_k}^T A_\mu A_\mu^T A_{\sigma'_k} A_{\sigma'_k}^T \ldots A_{\sigma'_1} A_{\sigma'_1}^T A_{\rho'} \right] \right\rangle \\
&= \left( \frac{N_s}{N_x} \right)^{k+k'} \left( \frac{N_x}{N_s} \right)^{k+k'+2} N_s \left( C_{\rho\sigma_1}^{in} C_{\sigma_1\sigma_2}^{in} \ldots C_{\sigma_k\mu}^{in} C_{\mu\sigma'_{k'}}^{in} \ldots C_{\sigma'_2\sigma'_1}^{in} C_{\sigma'_1\rho'}^{in} + \mathcal{O}(\frac{N_s}{N_x}) \right) \\
&= \frac{N_x^2}{N_s} \left( C_{\rho\sigma_1}^{in} C_{\sigma_1\sigma_2}^{in} \ldots C_{\sigma_k\mu}^{in} C_{\mu\sigma'_{k'}}^{in} \ldots C_{\sigma'_2\sigma'_1}^{in} C_{\sigma'_1\rho'}^{in} + \mathcal{O}\left( \frac{N_s}{N_x} \right) \right).
\end{aligned}
\tag{50}
$$

## B. Analysis of the Impact of Task Order on Continual Learning

### B.1. Linear Perturbation Analysis of the Order Dependence

To further investigate the order-dependence of the final error $\bar{\epsilon}_f$, we aim to decompose the error into interpretable features of task similarity matrix. To this end, we further constrain the input similarity matrix $C^{in}$ to

$$
C_{ij}^{in} = \begin{cases} 1 & (\text{if} \quad i = j) \\ m + \delta M_{ij} & (\text{otherwise}), \end{cases}
\tag{51}
$$

where $m$ is a constant satisfying $-1 < m < 1$. Here, $\delta M_{ij}$ is a small element-wise perturbation added in such a way that $C_{in}$ is a correlation matrix. We define an upper-triangular matrix that consists of the constant component as $\bar{M}$. $(i, j)$-th element of $\bar{M}$ takes $\bar{M}_{ij} = m$ if $j > i$, but $\bar{M}_{ij} = 0$ otherwise. This constant $\bar{M}$ assumption enables us to evaluate the effect of inverse matrix term in $\bar{\epsilon}_f$ analytically owing to the following lemma:

**Lemma B.1.** *For any $m$ satisfying $-1 < m < 1$, an upper-triangle matrix $\bar{M}$ satisfies*

$$
\left[ (I + \bar{M})^{-1} \right]_{ij} = \delta_{ij} - m[j > i]_+ (1 - m)^{j-i-1},
\tag{52}
$$

*where $[X]_+$ is the indicator function that returns 1 if $X$ is true, but returns 0 otherwise.*

*Proof.*

$$
\begin{aligned}
\left[ (I + \bar{M})(I + \bar{M})^{-1} \right]_{ij} &= \sum_{k=1}^{P} (\delta_{ik} + m[k > i]_+) \left( \delta_{kj} - m[j > k]_+ (1 - m)^{j-k-1} \right) \\
&= \delta_{ij} + [j > i]_+ \left( -m(1 - m)^{j-i-1} + m - m^2 \sum_{k=i+1}^{j-1} (1 - m)^{j-k-1} \right) \\
&= \delta_{ij} + [j > i]_+ \left( -m(1 - m)^{j-i-1} + m - m \left[ 1 - (1 - m)^{j-i-1} \right] \right) \\
&= \delta_{ij}.
\end{aligned}
\tag{53}
$$

$\square$

Let us consider the case when the output similarity is the same across all the tasks for simplicity. Then, $C^{out}$ is written as

$$
C^{out} = \rho_o \mathbf{1}\mathbf{1}^T + [1 - \rho_o] I.
\tag{54}
$$

In this problem setting, assuming that $\delta M$ is sufficiently small compared to $\bar{M}$, we can rewrite the error $\bar{\epsilon}_f$ as a linear function of $\delta M$, which enables us to interpret the contribution of different pairwise similarities to the final error. The following theorem describes the exact decomposition of the impact of task order in this linear perturbation limit.

**Theorem B.2.** *Let us suppose that all elements of a upper-triangle matrix with zero-diagonal components $\delta M_{ij}$ satisfies, $|\delta M_{ij}| < \delta_m$, where $\delta_m$ is a positive constant. Then, the error $\bar{\epsilon}_f$ is rewritten as below:*

$$\bar{\epsilon}_f = \bar{\epsilon}_f[m, \rho_o] + \sum_{i=1}^{P} \sum_{j=i+1}^{P} G_{ij} \delta M_{ij} + \mathcal{O}(\delta_m^2), \tag{55}$$

*where*

$$\bar{\epsilon}_f[m, \rho_o] \equiv \left\| \left(\rho_o \mathbf{1}\mathbf{1}^T + [1-\rho_o]I\right)^{1/2} (I + \bar{M})^{-1}\bar{M}^T \right\|_F^2 \tag{56a}$$

$$G_{ij} \equiv g_o(i) + g_o(j) + g_+(j+i) + g_-(j-i), \tag{56b}$$

*and functions $g_o, g_-, g_+ : \mathbb{Z} \to \mathbb{R}$ that constitute the coefficient $G$ are defined by*

$$g_o(k) \equiv \frac{(1-\rho_o)m}{2-m}(1-m)^{P-k} - \left(\rho_o - \frac{(1-\rho_o)m}{2-m}\right)(1-m)^{P+k-1}, \tag{57a}$$

$$g_+(s) \equiv \left(\rho_o - \frac{(1-\rho_o)m}{2-m}\right)\frac{3-m}{2-m}(1-m)^{s-1} - \frac{(1-\rho_o)m}{2-m}\left(Pm + 2(1-m) - \frac{(1-m)^2}{2-m}\right)(1-m)^{2P-s}, \tag{57b}$$

$$g_-(d) \equiv \frac{(1-\rho_o)m}{2-m}\frac{1}{2-m}(1-m)^{d-1} - \left[\left(\rho_o - \frac{(1-\rho_o)m}{2-m}\right)\left(1-(1-m)^P\left(\frac{mP}{1-m} + \frac{3-m}{2-m}\right)\right) - \frac{(1-\rho_o)m}{2-m}\frac{1}{1-m}\right](1-m)^{P-d}. \tag{57c}$$

Matrix $G$ specifies the relative contribution of each task-to-task similarity to the final error. Notably, $\bar{\epsilon}_f[m, \rho_o]$ term is permutation invariant by construction. Thus, task-order dependence stems from the $\delta M$-dependent terms.

Inserting $\rho_o = 1$ into Eqs. 57, you get Theorem 4.1 in the main text.

## B.2. Proof of Theorem B.2

The inverse of $(I + \bar{M} + \delta M)$ is rewritten as

$$(I + \bar{M} + \delta M)^{-1} = (I + \bar{M})^{-1} - (I + \bar{M})^{-1}\delta M(I + \bar{M})^{-1} + \mathcal{O}(\delta M^2). \tag{58}$$

Thus, up to the leading order with respect to $\delta M$, we have

$$\bar{\epsilon}_f = \left\| \left(\rho_o\mathbf{1}\mathbf{1}^T + [1-\rho_o]I\right)^{1/2}(I + \bar{M} + \delta M)^{-1}(\bar{M} + \delta M)^T \right\|_F^2$$

$$= \left\| \left(\rho_o\mathbf{1}\mathbf{1}^T + [1-\rho_o]I\right)^{1/2}(I + \bar{M})^{-1}\bar{M}^T \right\|_F^2$$

$$+ 2\mathrm{tr}\left[\bar{M}(I + \bar{M})^{-T}\left(C^{out}\right)^{1/2}(I + \bar{M})^{-1}\left(\delta M^T - \delta M(I + \bar{M})^{-1}\bar{M}^T\right)\right] + \mathcal{O}(\delta M^2). \tag{59}$$

The first term corresponds to $\bar{\epsilon}_f[\bar{M}, C^{out}]$, thus it is enough to show that the coefficients of $\delta M$ is written as Eqs. 56b and 57.

$\bar{M}(I + \bar{M})^{-T}$ term is rewritten as

$$[\bar{M}(I + \bar{M})^{-T}]_{ij} = \sum_{k=1}^{P} m[k > i]_+ \left(\delta_{jk} - m[k > j]_+(1-m)^{k-j-1}\right)$$

$$= m[j > i]_+ - m\left((1-m)^{k_{ij}-j-1} - (1-m)^{P-j}\right)$$

$$= [j > i]_+ m\left(1 - [1 - (1-m)^{P-j}]\right) - m[j \leq i]_+\left((1-m)^{i-j} - (1-m)^{P-j}\right)$$

$$= m(1-m)^{P-j} - m[j \leq i]_+(1-m)^{i-j}. \tag{60}$$

In the second line, we defined $k_{ji}$ by $k_{ji} \equiv \max(i+1, j+1)$. For the ease of notation, let us denote the last term in the equation above as

$$v_{ji} \equiv m(1-m)^{P-j} - m[j \leq i]_+(1-m)^{i-j}. \tag{61}$$

Next, $\left(\rho_o \mathbf{1}\mathbf{1}^T + [I - \rho_o]I\right)(I + \bar{M})^{-1}$ term becomes

$$
\begin{aligned}
&\left[\left(\rho_o \mathbf{1}\mathbf{1}^T + [I - \rho_o]I\right)(I + \bar{M})^{-1}\right]_{ij} \\
&= \sum_k (\rho_o + (1 - \rho_o)\delta_{ik})\left(\delta_{kj} - m[j > k]_+(1 - m)^{j-k-1}\right) \\
&= \rho_o\left(1 - \sum_k m[j > k]_+(1 - m)^{j-k-1}\right) + (1 - \rho_o)\left(\delta_{ij} - m[j > i]_+(1 - m)^{j-i-1}\right) \\
&= \rho_o(1 - m)^{j-1} + (1 - \rho_o)\delta_{ij} - (1 - \rho_o)m[j > i]_+(1 - m)^{j-i-1}
\end{aligned}
\tag{62}
$$

In the last line, we used $\sum_k m[j > k]_+(1 - m)^{j-k-1} = 1 - (1 - m)^{j-1}$. As before, let us denote the coefficient by

$$
u_{ij} \equiv \rho_o(1 - m)^{j-1} - (1 - \rho_o)m[j > i]_+(1 - m)^{j-i-1}.
\tag{63}
$$

Then, the first-order term with respect to $\delta M$ follows

$$
\begin{aligned}
&\mathrm{tr}\left[\bar{M}(I + \bar{M})^{-T}\left(\rho_o \mathbf{1}\mathbf{1}^T + (1 - \rho_o)I\right)(I + \bar{M})^{-1}\left(\delta M^T - \delta M(I + \bar{M})^{-1}\bar{M}^T\right)\right] \\
&= \sum_{ijk}\left[\bar{M}(I + \bar{M})^{-T}\right]_{ij}\left[\left(\rho_o \mathbf{1}\mathbf{1}^T + [I - \rho_o]I\right)(I + \bar{M})^{-1}\right]_{jk}\left(\delta M_{ik} - \sum_l \delta M_{kl}\left[\bar{M}(I + \bar{M})^{-T}\right]_{il}\right) \\
&= \sum_{ijk} v_{ji}\left((1 - \rho_o)\delta_{jk} + u_{jk}\right)\left(\delta M_{ik} - \sum_l \delta M_{kl}v_{li}\right) \\
&= \sum_{kl} \delta M_{kl}\sum_{ij} v_{ji}\left(\delta_{ik}[(1 - \rho_o)\delta_{jl} + u_{jl}] - v_{li}[(1 - \rho_o)\delta_{jk} + u_{jk}]\right) \\
&= \sum_{kl} \delta M_{kl}G_{kl},
\end{aligned}
\tag{64}
$$

where the coefficient of $(k, l)$-th element is defined by

$$
G_{kl} \equiv \sum_{ij} v_{ji}\left(\delta_{ik}[(1 - \rho_o)\delta_{jl} + u_{jl}] - v_{li}[(1 - \rho_o)\delta_{jk} + u_{jk}]\right).
\tag{65}
$$

$G_{kl}$ is decomposed into

$$
G_{kl} = (1 - \rho_o)v_{lk} + \sum_j v_{jk}u_{jl} - (1 - \rho_o)\sum_i v_{ki}v_{li} - \sum_{ij} v_{ji}v_{li}u_{jk}.
\tag{66}
$$

The first term $(1 - \rho_o)v_{lk}$ is rewritten as

$$
\begin{aligned}
(1 - \rho_o)v_{lk} &= (1 - \rho_o)m(1 - m)^{P-l} - m[l \le k]_+(1 - m)^{k-l} \\
&= (1 - \rho_o)m(1 - m)^{P-l}.
\end{aligned}
\tag{67}
$$

Here, we dropped the second term, because $\delta M_{kl} = 0$ when $l \le k$.

Regarding the second term, summation over $j$ is evaluated as

$$
\sum_{j=1}^{P} v_{ji}u_{jk} = \sum_{j} \left( m(1-m)^{P-j} - m[j \le i]_+(1-m)^{i-j} \right) \left( \rho_o(1-m)^{k-1} - (1-\rho_o)m[k>j]_+(1-m)^{k-j-1} \right)
$$

$$
= \rho_o(1-m)^{k-1} \left( m\sum_{j=1}^{P}(1-m)^{P-j} - m\sum_{j=1}^{i}(1-m)^{i-j} \right)
$$

$$
- (1-\rho_o)m^2 \left( \sum_{j=1}^{k-1}(1-m)^{(P-j)+(k-j-1)} - \sum_{j=1}^{j_{ik}}(1-m)^{(i-j)+(k-j-1)} \right)
$$

$$
= \rho_o(1-m)^{k-1}\left( (1-m)^i - (1-m)^P \right)
$$

$$
- \frac{(1-\rho_o)m}{2-m}\left( \left[(1-m)^{P-(k-1)} - (1-m)^{P+(k-1)}\right] - \left[(1-m)^{i+k-1-2j_{ik}} - (1-m)^{i+k-1}\right] \right)
$$

$$
= \left( \rho_o - \frac{(1-\rho_o)m}{2-m} \right)\left( (1-m)^{i+k-1} - (1-m)^{P+k-1} \right) + \frac{(1-\rho_o)m}{2-m}\left( (1-m)^{i+k-1-2j_{ik}} - (1-m)^{P-(k-1)} \right).
\tag{68}
$$

In the third line, we defined $j_{ik}$ as $j_{ik} \equiv \min(k-1, i)$. Thus, the second, $\sum_j v_{jk}u_{jl}$, term becomes

$$
\sum_{j} v_{jk}u_{jl} = -\left( \left[\rho_o - \frac{(1-\rho_o)m}{2-m}\right](1-m)^{P+l-1} + \frac{(1-\rho_o)m}{2-m}(1-m)^{P-l+1} \right)
$$

$$
+ \left[\rho_o - \frac{(1-\rho_o)m}{2-m}\right](1-m)^{k+l-1} + \frac{(1-\rho_o)m}{2-m}(1-m)^{l-k-1}.
\tag{69}
$$

The third term is, from a similar calculation, rewritten as

$$
\sum_{i} v_{li}v_{ki}
$$

$$
= \sum_{i} \left( m(1-m)^{P-l} - m[l \le i]_+(1-m)^{i-l} \right)\left( m(1-m)^{P-k} - m[k \le i]_+(1-m)^{i-k} \right)
$$

$$
= Pm^2(1-m)^{2P-(k+l)} - m^2\left( (1-m)^{P-k}\sum_{i=l}^{P}(1-m)^{i-l} + (1-m)^{P-l}\sum_{i=k}^{P}(1-m)^{i-k} \right) + m^2\sum_{i=l}^{P}(1-m)^{2i-(k+l)}
$$

$$
= \left( Pm^2 + 2m(1-m) - \frac{m(1-m)^2}{2-m} \right)(1-m)^{2P-(k+l)} - m\left( (1-m)^{P-k} + (1-m)^{P-l} \right) + \frac{m}{2-m}(1-m)^{l-k}.
\tag{70}
$$

The last term is a little more complicated, so let us divide it into two terms:

$$
\sum_{i} v_{li}\sum_{j} v_{ji}u_{jk} = \left( \rho_o - \frac{(1-\rho_o)m}{2-m} \right)T_1 + \frac{(1-\rho_o)m}{2-m}T_2,
\tag{71}
$$

where

$$
T_1 = m(1-m)^{P-l}\sum_{i=1}^{P}\left( (1-m)^{i+k-1} - (1-m)^{P+k-1} \right) - m\sum_{i=l}^{P}(1-m)^{i-l}\left( (1-m)^{i+k-1} - (1-m)^{P+k-1} \right),
$$

$$
T_2 = m(1-m)^{P-l}\sum_{i=1}^{P}\left( (1-m)^{i+k-1-2j_{ik}} - (1-m)^{P-(k-1)} \right) - m\sum_{i=l}^{P}(1-m)^{i-l}\left( (1-m)^{i-(k-1)} - (1-m)^{P-(k-1)} \right).
\tag{72a}
$$

Taking summation over index $i$, $T_1$ and $T_2$ are rewritten as

$$T_1 = m(1-m)^{P-l} \left( \frac{(1-m)^k}{m} \left[ 1 - (1-m)^P \right] - P(1-m)^{P+k-1} \right)$$

$$- \left( \frac{1}{2-m} \left[ (1-m)^{l+k-1} - (1-m)^{2P+1+(k-l)} \right] - (1-m)^{P+k-1} \left[ 1 - (1-m)^{P+1-l} \right] \right)$$

$$= \left( 1 - (1-m)^P \left[ 2 + \frac{mP}{1-m} - \frac{1-m}{2-m} \right] \right) (1-m)^{P-(l-k)} - \frac{1}{2-m} (1-m)^{l+k-1} + (1-m)^{P+k-1}. \quad (73)$$

$$T_2 = (1-m)^{P-l} \left( [1 - (1-m)^{k-1}] + [(1-m) - (1-m)^{P+2-k}] - mP(1-m)^{P-(k-1)} \right)$$

$$- \left( \frac{1}{2-m} [(1-m)^{l-k+1} - (1-m)^{2P+3-(l+k)}] - [1 - (1-m)^{P+1-l}](1-m)^{P-(k-1)} \right)$$

$$= -\frac{1-m}{2-m} (1-m)^{l-k} - \frac{1}{1-m}(1-m)^{P-(l-k)} + \left( (2-m)(1-m)^{P-l} + (1-m)^{P-k+1} \right)$$

$$- (1-m)^{2P-(k+l)} \left( mP(1-m) + 2(1-m)^2 - \frac{(1-m)^3}{2-m} \right). \quad (74)$$

The results above show that $G_{kl}$ is decomposed into four components:

$$G_{kl} = g_-(l-k) + g_+(l+k) + g_L(k) + g_R(l). \quad (75)$$

Summing over the terms that only depends on $k$, we have

$$g_L(k) = -(1-\rho_o) \left( -m(1-m)^{P-k} \right) - \left( \rho_o - \frac{(1-\rho_o)m}{2-m} \right) (1-m)^{P+k-1} - \frac{(1-\rho_o)m}{2-m}(1-m)^{P-k+1}$$

$$= \frac{(1-\rho_o)m}{2-m}(1-m)^{P-k} - \left( \rho_o - \frac{(1-\rho_o)m}{2-m} \right)(1-m)^{P+k-1} \quad (76)$$

Similarly, the terms that only depend on $l$ are summed up to:

$$g_R(l) = (1-\rho_o)m(1-m)^{P-l} - \left( \left[ \rho_o - \frac{(1-\rho_o)m}{2-m} \right] (1-m)^{P+l-1} + \frac{(1-\rho_o)m}{2-m}(1-m)^{P-l+1} \right)$$

$$- (1-\rho_o) \left( -m(1-m)^{P-l} \right) - \frac{(1-\rho_o)m}{2-m}(2-m)(1-m)^{P-l}$$

$$= \frac{(1-\rho_o)m}{2-m}(1-m)^{P-l} - \left( \rho_o - \frac{(1-\rho_o)m}{2-m} \right)(1-m)^{P+l-1}. \quad (77)$$

Therefore, $g_L$ and $g_R$ have the same form. We denote this function as $g_o$ below.

$g_+(l+k)$ term has a slightly more complicated expression:

$$g_+(l+k) = \left( \rho_o - \frac{(1-\rho_o)m}{2-m} \right)(1-m)^{k+l-1} - (1-\rho_o)m \left( Pm + 2(1-m) - \frac{(1-m)^2}{2-m} \right)(1-m)^{2P-(k+l)}$$

$$+ \left( \rho_o - \frac{(1-\rho_o)m}{2-m} \right) \frac{1}{2-m}(1-m)^{l+k-1} + \frac{(1-\rho_o)m}{2-m}(1-m)\left( Pm + 2(1-m) - \frac{(1-m)^2}{2-m} \right)(1-m)^{2P-(k+l)}$$

$$= \left( \rho_o - \frac{(1-\rho_o)m}{2-m} \right)\frac{3-m}{2-m}(1-m)^{l+k-1} - \frac{(1-\rho_o)m}{2-m} \left( Pm + 2(1-m) - \frac{(1-m)^2}{2-m} \right)(1-m)^{2P-(k+l)}. \quad (78)$$

Lastly, $g_-(l-k)$ term becomes

$$g_-(l-k)$$

$$= \frac{(1-\rho_o)m}{2-m}(1-m)^{l-k-1} - \frac{(1-\rho_o)m}{2-m}(1-m)^{l-k} - \left( \rho_o - \frac{(1-\rho_o)m}{2-m} \right) \left( 1 - (1-m)^P \left( 2 + \frac{mP}{1-m} - \frac{1-m}{2-m} \right) \right)(1-m)^{P-(l-k)}$$

$$+ \frac{(1-\rho_o)m}{2-m} \left( \frac{1-m}{2-m}(1-m)^{l-k} + \frac{1}{1-m}(1-m)^{P-(l-k)} \right)$$

$$= \frac{(1-\rho_o)m}{2-m}\frac{1}{2-m}(1-m)^{l-k-1} - \left[ \left( 1 - (1-m)^P \left( \frac{mP}{1-m} + \frac{3-m}{2-m} \right) \right) - \frac{(1-\rho_o)m}{2-m}\frac{1}{1-m} \right](1-m)^{P-(l-k)}. \quad (79)$$

We thus obtain Eq. 57. If we set $\rho_o = 1$, we recover Theorem 4.1. In Theorem 4.1, $G^+$ is defined as $G_{ij}^+ = g_+(i+j) + g_L(i) + g_R(j)$.

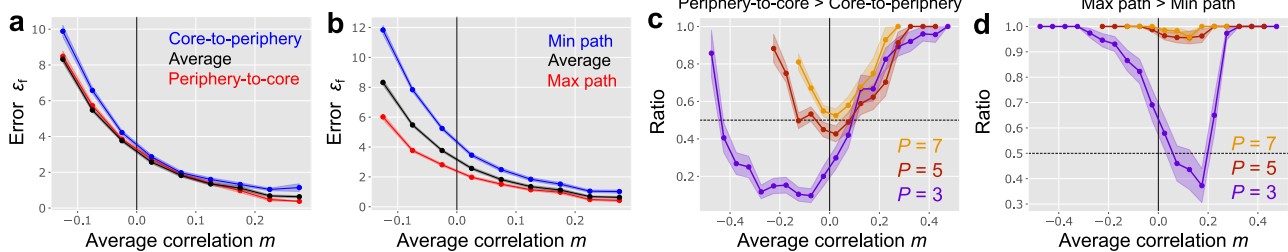

*Figure 6.* **a, b)** The same as Fig. 3e and g, but without normalization. The average correlation dominates the error $\epsilon_f$, but the order dependence is observed robustly. **c)** The ratio of networks where the periphery-to-core order outperforms core-to-periphery order. As before, we generated 1000 samples of $C^{in}$ matrices randomly while fixing $C^{out}$ to be all one, then binned $C^{in}$ based on the average off-diagonal correlation. For each average correlation value $m$, we then calculated the ratio of networks in which the periphery-to-core order achieved smaller error than the core-to-periphery order. **d)** The same as panel c, but the ratio of random seeds where the max-path orders achieved the better performance than the min-path orders.

### B.3. The Impact of Task Typicality

We can further analyze the impact of task typicality on the task order discuss in the main text. Let us define $\bar{g}$ by

$$\bar{g} \equiv \frac{1}{P} \sum_{\mu=1}^{P} (1-m)^{\mu} = \frac{1}{mP} \left( (1-m) - (1-m)^{P+1} \right), \tag{80}$$

then $G_{\mu\nu}^{+}$ is rewritten as

$$G_{\mu\nu}^{+} = \left[ \frac{(3-m)\bar{g}}{(2-m)(1-m)} - (1-m)^{P-1} \right] \left[ (1-m)^{\mu} + (1-m)^{\nu} \right]$$
$$+ \frac{3-m}{(2-m)(1-m)} \left( (1-m)^{\mu} - \bar{g} \right) \left( (1-m)^{\nu} - \bar{g} \right) - \frac{(3-m)\bar{g}^2}{(2-m)(1-m)}. \tag{81}$$

Thus, the corresponding error term $\delta\bar{\epsilon}_f^{+}$ becomes

$$\delta\bar{\epsilon}_f^{+} = \sum_{\mu=1}^{P} \sum_{\nu=\mu+1}^{P} G_{\mu\nu}^{+} \delta M_{\mu\nu}$$
$$= P \left[ \frac{(3-m)\bar{g}}{(2-m)(1-m)} - (1-m)^{P-1} \right] \sum_{\mu=1}^{P} (1-m)^{\mu} \delta t_{\mu}$$
$$+ \frac{3-m}{(2-m)(1-m)} \sum_{\mu=1}^{P} \sum_{\nu=\mu+1}^{P} \left( (1-m)^{\mu} - \bar{g} \right) \left( (1-m)^{\nu} - \bar{g} \right) \delta M_{\mu\nu} + \text{const.} \tag{82}$$

If $\frac{(3-m)\bar{g}}{(2-m)(1-m)} > (1-m)^{P-1}$, from the rearrangement inequality, the first term is minimized under $\delta t_1 \leq \delta t_2 \leq ... \leq \delta t_P$ ordering. However, the second term may not be minimized under this task order.

## C. Implementation Details

Source codes for all the numerical experiments are available at https://github.com/ziyan-li-code/optimal-learn-order.

### C.1. Linear Teacher-student Model with Latent Variables

In the simulations depicted in Figs. 2b, 3eg, and 4ef, we set the latent vector size $N_s = 30$, input layer size $N_x = 3000$, and the output layer size $N_y = 10$. We initialized the weight matrix $W$ as the zero matrix, and then updated the weight using gradient descent (Eq. 21) with learning rate $\eta = 0.001$ for 100 iterations per task. In Fig. 2b, the input correlation matrix $C^{in} \in \mathbb{R}^{P \times P}$ was generated randomly in the following manner. First, we generated a strictly upper-triangular matrix

*Figure 7.* Task order preference in continuous image classification tasks. **a,b)** Average classification performance after continual learning of CIFAR-100 where each task consists of 5 label classifications. Here, we randomly picked 25 labels from CIFAR-100 dataset and generated 5 tasks each requiring classification of 5 labels. **c,d)** Average classification performance after continual learning on CIFAR-10. Task similarity was estimated by evaluating zero-shot generalization performance from task $A$ to $B$, using the training data of $B$ instead of the test data (see Appendix C.4 for details).

$C^{in,U}$ by sampling each element independently from a continuous uniform distribution between 0 and 1, $\mathcal{U}_{[0,1]}$. We then generated the full matrix $C^{in}$ by $C^{in} = C^{in,U} + (C^{in,U})^T + I$. If the resultant $C^{in}$ is a positive semi-definite matrix, we accepted the matrix, otherwise, we generated $C^{in}$ in the same manner, until we obtain a positive semi-definite matrix. Here, we limited the correlation to be positive mainly because continual learning is typically impractical when there exists a large negative correlation between tasks. We generated $C^{out}$ using the same method. In Fig. 2c, we estimated the final error $\bar{\epsilon}_f$ using Eq. 8 for each triplet $(\rho_{AB}, \rho_{BC}, \rho_{CA})$. We calculated the error for all six task orders and plotted the order that yielded the minimum error.

In Figs. 3e and g, we generated input correlation matrix $C^{in}$ using the same method with Fig. 2b, but we instead sampled the elements from a uniform distribution between -1 and 1, $\mathcal{U}_{[-1,1]}$. The average correlation $m$ was defined by the average of the off-diagonal components of $C^{in}$. The output correlation matrix $C^{out}$ was set to be the all one matrix (i.e. $C^{out}_{\mu\nu} = 1$ for all $(\mu, \nu)$ pairs) which corresponds to the $\rho_o = 1$ scenario. We estimated the error under each task order for 1000 randomly generated input correlation matrices and binned the performance by the average correlation. The average performance (black lines in Figs. 3e and g) were estimated by taking the average over randomly sampled 100 task orders. The error bars, representing the standard error of mean, are larger for larger average correlation because we didn't generated many $C^{in}$ with a large average correlation under our random generation method. Because there are two task sequence that provides the max-path due to symmetry (e.g. $A{\to}C{\to}E{\to}B{\to}D$ and $D{\to}B{\to}E{\to}C{\to}A$ in Fig. 3c), we defined the error of the max-path rule as the average over these two task orders.

### C.2. Generation of Input Similarity Matrices Having Simple Graph Structures

In Fig. 4, we introduced simple graph structures to the task similarity matrices. To this end, we first generated an unweighted bidirectional adjacency matrix $A_{dj} \in \{0,1\}^{P \times P}$ given a graph structure, then calculated distance between nodes on the graph specified by $A_{dj}$, which we denote as $D$. From this distance matrix $D$, we generated the input similarity matrix $C^{in}$ by $C^{in}_{ij} = a^{D_{ij}}$ for each task pairs $(i, j)$. Here, $a$ is the constant that controls overall task similarity. $a \approx 1$ means that inputs for all tasks are highly correlated, whereas $a \approx 0$ means that they are mostly independent.

For instance, in the case of chain graph, the adjacency matrix is given by $A_{dj,ij} = 1$ if $j = i \pm 1$ else $A_{dj,ij} = 0$. Thus, the distance between nodes $D$ follows $D_{ij} = |i - j|$ and the input correlation matrix becomes $C^{in}_{ij} = a^{|i-j|}$. In the ring graph, the distance between node is instead given by $D_{ij} = \min\{|i - j|, P - |i - j|\}$. For the tree graph, we used a tree where each non-leaf node has exactly two children nodes. The same structure was assumed for the leaves graph, except that we only used leaf nodes for constructing the task similarity matrix.

### C.3. Convolutional and Multi-layered Non-linear Neural Networks for Image Classification

We used convolutional neural networks (CNNs) for numerical experiments with the CIFAR-10 and CIFAR-100 datasets. The network consisted of two convolutional layers and one dense layer, followed by an output layer. The first convolutional layer had 32 filters with $3 \times 3$ kernels. The output was passed through a Rectified Linear Unit (ReLU) activation function and then downsampled using average pooling with a window size of $2 \times 2$ and a stride of 2. The second convolutional layer was similar to the first, except that we used 64 filters. The dense layer following the two convolutional layers had 256

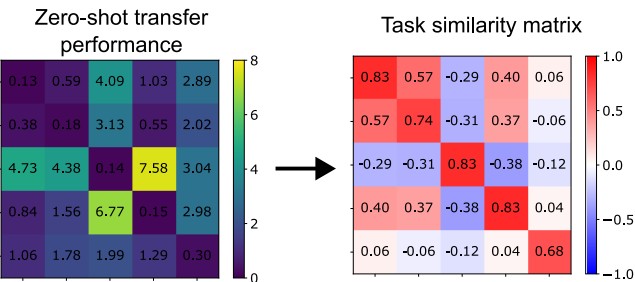

*Figure 8.* Schematic of the task similarity estimation. From the error in zero-shot transfer $\epsilon_\mu[W_\nu]$ (left), we estimated task similarity $\rho_{\mu\nu}$ (right).

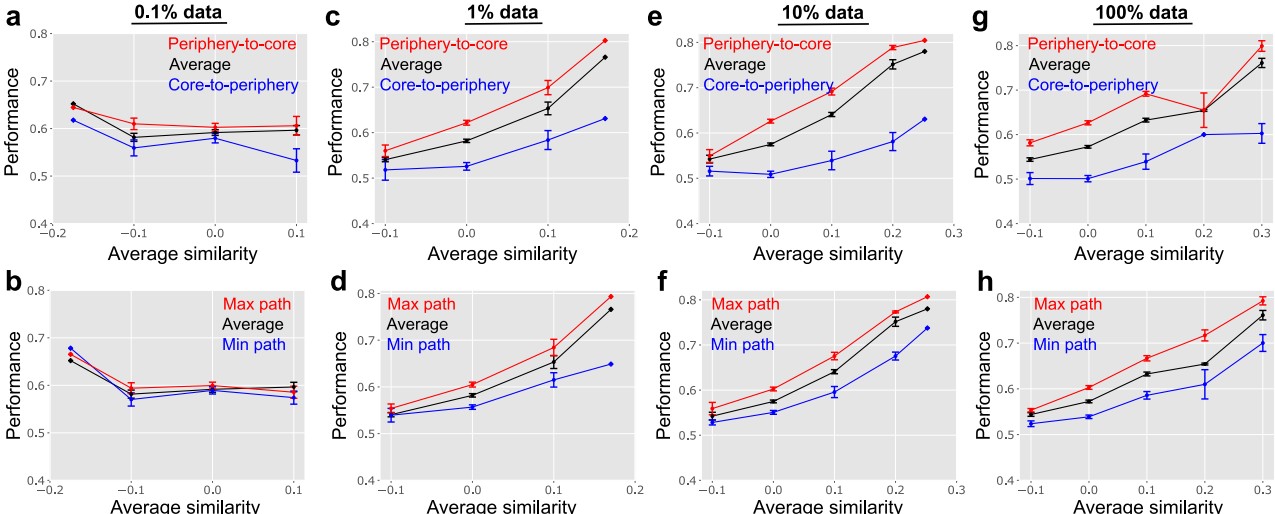

*Figure 9.* Task order preference estimated from various fraction of training data under CIFAR-10 task. For instance, in $0.1\%$ data results depicted in panels a and b, we first used $0.1\%$ of training data ($\sim 10$ images) for training the network with one task and used $0.1\%$ of training data from another task to evaluate the zero-shot transfer performance. We then estimated the task similarity as before. Panels g and h are the same with panels c and d in Fig. 7. We used the same set of parameters with the results depicted in the main figure (see Appendix C for details).

neurons, with ReLU as the activation function. For classification, we used a softmax activation function in the last layer. The weights of both convolutional and dense layers were initialized with LeCun normal initializers.

For the Fashion-MNIST dataset, we used multi-layer perceptrons (MLPs) to evaluate the robustness of our findings against the neural architecture. The MLP model had two hidden layers with 128 and 64 neurons, respectively. We used ReLU as the activation function for the hidden layers and softmax for the output layer.

In both CNN and MLP models, we studied binary classification with two output neurons, except in Figure 7a and b, where we considered a classification of 5 labels with five output neurons. All tasks were implemented as single-head continual learning where the output nodes are shared across tasks. The performance was evaluated by the average classification accuracy on the test datasets for all the tasks at the end of the entire training.

The networks were trained by minimizing the cross-entropy loss using the Adam optimizer with a learning rate of $10^{-3}$ for five epochs per task. We set the batch size to 4 due to GPU memory constraints. The models were implemented using Flax (Heek et al., 2024), a JAX neural network library, and were trained on NVIDIA Tesla V100 GPUs

In both Figures 5 and 7, we generated 100 task sets by randomly dividing 10 labels into a set of five binary classification tasks. In the case of CIFAR-100, we initially sampled 10 labels randomly, then partitioned them into five binary classification. We binned the task sets by the average similarity estimated from Eq. 16, then plotted the mean performance and the standard error of mean for each bin. The black lines representing the average performance of random task order were estimated by

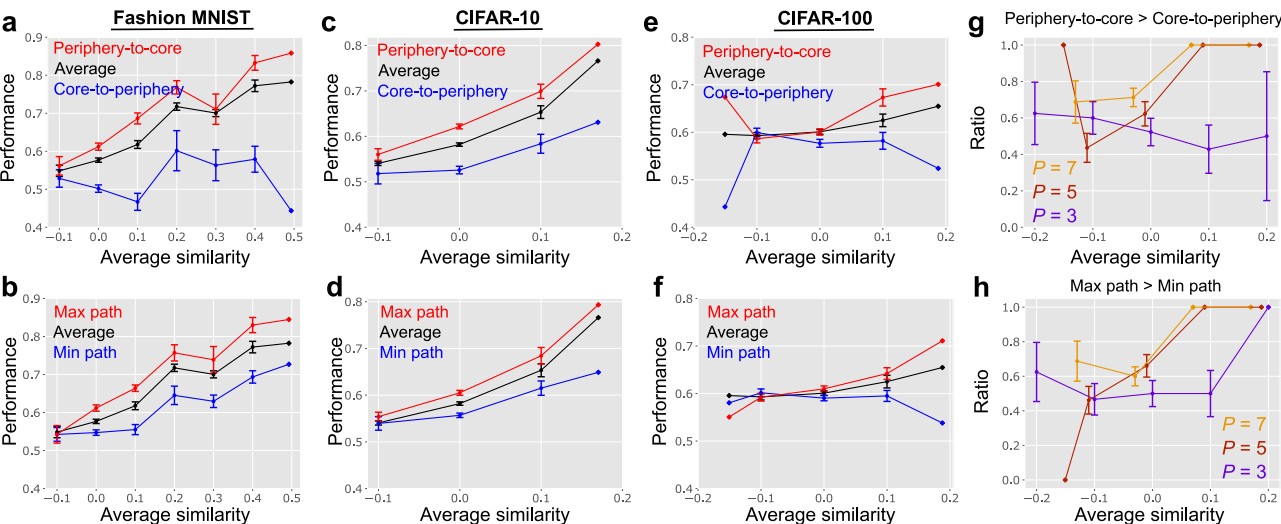

*Figure 10.* Task order preference in continuous image classification tasks where the task similarity was estimated from 1% of training data, as opposed to Fig. 5 where 100% of data was used. **a–f)** Continual learning performance, defined as the average test accuracy across all the tasks after learning, under various task orders. **g, h)** The ratio of task sets where the periphery-to-core rule outperforms the core-to-periphery rule (g), and where the max-path rule outperforms the min-path rule (h), under CIFAR-100 with different numbers of tasks ($P = 3, 5, 7$). Panels c and d were replicated from Fig. 9 for completeness. Note that task-order effects were smaller in CIFAR-100 because the number of training images per label is fewer in CIFAR-100 than in CIFAR-10 and Fashion-MNIST.

taking the mean of the classification performance under 30 random task orders for each task set. Because there are two task orders that provides the max-path by construction, we define the performance of the max-path rule as the mean of performance under these two task orders.

## C.4. Estimation of Task Similarity

In the main text, we inferred similarity between two tasks *A* and *B* using zero-shot transfer performance between the two tasks. Previous work on linear model indicates that, if the output similarity is one, the pairwise transfer performance $\Delta \epsilon_{TF}[\nu \to \mu] \equiv \epsilon_\mu[W_\nu] - \epsilon_\mu[W_o]$ is written as (Hiratani)

$$\Delta \epsilon_{TF}[\nu \to \mu] = \rho^{in}_{\mu\nu}(2 - \rho^{in}_{\mu\nu}), \tag{83}$$

indicating that the input similarity between two tasks can be inferred as

$$\rho^{in}_{\mu\nu} = 1 - \sqrt{1 - \Delta \epsilon_{TF}[\nu \to \mu]}. \tag{84}$$

Motivated by this relationship, we defined similarity between tasks in general nonlinear networks by Eq. 16. A similar approach was implemented in (Lad et al., 2009). Notably, this method only requires inputs/outputs of the trained network, and thus applicable to situation where the model details are inaccessible (e.g., human and animal brains, closed-LLM).

We implemented the evaluation the zero-shot transfer performance from task *A* to *B* as follows: First, we trained a network on task *A* for 5 epochs from a random initialization using the Adam optimizer on the cross-entropy loss with learning rate $10^{-3}$ as above. We then measured the cross-entropy loss on the test dataset of task *B*, $\epsilon_B[W_A]$, where $W_A$ represents the weight after 5 epochs of training on task *A*. To normalize the accuracy, we divided the obtained loss by the loss on task *B* under a label shuffling, $\epsilon_{B,sf}[W_A]$. The resultant value $\frac{\epsilon_B[W_A]}{\epsilon_{B,sf}[W_A]}$ characterizes how well the network transfer to task *B* compared to a random task with the same input statistics. Since evaluating similarity using the transfer performance on the test data may potentially introduce bias, in Fig. 7c and d, we estimated the transfer performance $\Delta \epsilon_B[W_A]$ using the training dataset for task *B*. Even in this setting, we found results nearly identical to those in Fig. 5c and d, confirming the robustness of our findings with respect to the details of the similarity evaluation method.

In the task similarity and order estimation from sparse data shown in Figs. 9 and 10, we estimated task similarity in the same manner as described above, but using subsampled training data. For example, in the 1% data scenario, we used 1% of

the training data for task $A$ to calculate the weights after learning task $A$, denoted as $W_A$. As before, we trained the network using the Adam optimizer on the cross-entropy loss for 5 epochs from random initialization except that we only used 1% of training data at each epoch. We then estimated the transfer performance to task $B$, $\epsilon_B[W_A]$, using 1% of the training data for task $B$. The task similarity between tasks $A$ and $B$ was then computed using Eq. 16, as in the full data setting.

