# OpenReview forum: "Optimal Task Order for Continual Learning of Multiple Tasks"
_ICML.cc/2025/Conference — ICML 2025 poster_

### Official Review · Reviewer_X6mc · 2025-02-17

**Overall Recommendation:** 3

**Summary:**

This paper analyzed the impact of task order and task similarity on continual learning. With a linear teacher-student model, the authors derived an analytical expression that connects task order and task similarity to the learning performance. The theoretical results further derive two principles (the periphery-to-core rule and the max-path rule), validated by extensive experiments.

## Update After Rebuttal

I thank the authors for their rebuttal. It addressed some of my concerns. I think the concerns about the use of task order has been partially addressed, where the proposed insight is applicable to the scenario with a small amount of available data rather than that without any multi-task data. As also mentioned by other reviewers, the real-world applicability remains a challenge. Overall, I appreciate the theoretical efforts and the interesting findings in this paper. I think it's valuable to better understand the continual learning problem. Therefore, I keep my rating toward accepting this paper.

**Claims And Evidence:**

Both theoretical and empirical results are presented clearly and convincing.

**Essential References Not Discussed:**

N/A

**Experimental Designs Or Analyses:**

The experimental results are extensive and insightful.

**Methods And Evaluation Criteria:**

The proposed method is reasonable to me.

**Other Comments Or Suggestions:**

N/A

**Other Strengths And Weaknesses:**

Strengths:

1. This paper is well organized and easy to follow.

2. The analytic expression is very interesting. It provides a clear connections of task order and task similarity to continual learning, which has been empirically investigated in literature yet without strong theoretical foundations.

3. The theoretical results, especially for the proposed two principles, are validated by extensive experiments.

Weaknesses:

I appreciate most of the contributions in this paper. However, there remains some minor concerns about applicability of the theoretical results.

1. This paper focused on the impact of task order and task similarity on continual learning. It's good to have more in-depth theoretical insights into these critical elements. However, I'm not sure the practical benefits of using these insights to improve continual learning, since the task order in continual learning usually cannot be known and controlled in advance (or else it should be curriculum learning). As a result, the proposed two principles may not provide practical benefits in applications.

2. As acknowledged by the authors, the entire framework is based on task-incremental learning, one of the most basic continual learning settings. Is it possible to extend the theoretical results to other continual learning settings, such as class-incremental learning? Would the impact of task order and task similarity change?

3. The authors claimed that they validate the theoretical results on real-world image classification datasets (Fashion-MNIST, CIFAR-10, CIFAR-100). I think CIFAR-10/100 is still relatively simple and well-crafted, rather than "real-world". I'm curious if the proposed principles are applicable to larger-scale images, such as ImageNet, but it's not a big issue.

**Questions For Authors:**

Please refer to the Weaknesses.

**Relation To Broader Scientific Literature:**

N/A

**Theoretical Claims:**

I check the main results, but not all of the details.

---

> ### Author Rebuttal · Authors · 2025-03-31
>
> Thank you for your helpful comments. Please find our replies below.
>
> > However, I'm not sure the practical benefits of using these insights to improve continual learning
>
> To illustrate the practical applicability of our work, let us first note that, our method requires only a small number of data samples to estimate the optimal task order.
> We newly confirmed it by estimating the task similarity and the optimal order using a small subset of training data in our split CIFAR-10 experiment.
>
> Table attached to the reply to Reviewer Z3Hz describes the average performance of various task orders on split CIFAR-10, where task similarity and the resulting order are estimated from a small subset of data.
> We found that, even when using only 1\% of the data, the max-path and periphery-to-core rules consistently outperformed the random task order (please see the explanation below the table for details).
>
> This result suggests that task-order optimization is feasible even with limited observations, opening up many practical applications.
> For instance, let us consider training a self-driving algorithm across three distinct environments: a suburban area in Phoenix, the city center of Tokyo, and the mountainous regions of the Swiss Alps. Due to geographical and economic constraints, it may be impractical to conduct simultaneous multi-task training across all three locations. While offline multi-task learning is an option, it requires full data acquisition beforehand. An alternative approach is to run small-scale pilot experiments in each environment, measure task similarity, and then determine the most effective order for conducting full-scale training.
> Please also note that this problem is somewhat different from conventional curriculum learning settings because the goal is to achieve high performance in all three environments.
>
> A similar challenge arises in the context of multi-task learning in robotics. In real-world settings, switching between tasks often involves physically rearranging objects around the robot, which is both time-consuming and labor-intensive. As a result, switching tasks on a trial-by-trial basis is often impractical, necessitating block-wise training instead. In such scenarios, optimizing task order could help maximize average performance across all tasks.
>
> Another long-term goal is school curriculum design, in which students aim to master diverse subjects while learning them sequentially. Our work may generalize to this problem, given the parallels between artificial and biological neural networks.
>
> We thus believe there are many potential applications for our theoretical framework.
>
> > Is it possible to extend the theoretical results to other continual learning settings, such as class-incremental learning?
>
> Thank you for raising an interesting point. Our latent model formulation is indeed applicable to class-incremental setting, unlike traditional linear model of continual learning, which cannot address class-incremental setting (as different classes do not interfere by definition).
>
> Let us focus on classification problems. In this context, the task-incremental setting corresponds to the case where $C^{\text{out}}$ is an all-ones matrix (i.e., $\rho_o = 1$), since all tasks share the same set of output labels. This motivates the analysis of the $\rho_o = 1$ case presented in Section 4.
>
> In contrast, class-incremental learning corresponds to the case where $C^{out}$ is the identity matrix (i.e., $\rho_o = 0$). This reflects the fact that, in the class-incremental setting, tasks do not share any labels, resulting in zero output correlation across tasks.
> At $\rho_o = 0$, the dependence of $G_{\mu\nu}^+$ and $G_{\mu\nu}^-$ on $\mu$ and $\nu$ differs from that in the $\rho_o = 1$ case (see Eq. 57), leading to a different task order preference. In particular, under $\rho_o = 0$, $G^+\_{\mu\nu}$ is no longer a monotonically decreasing function of $\mu$ and $\nu$, and thus the periphery-to-core rule is no longer a reliable heuristic.
> On the other hand, $G^-_{\mu\nu}$ approximately follows the form $\frac{m}{(2 - m)^2} (1 - m)^{\nu - \mu - 1}$, suggesting that, when $m > 0$, tasks that are temporally nearby should be dissimilar in order to minimize error. This is because $\frac{m}{(2 - m)^2} (1 - m)^{d - 1}$ is a monotonically decreasing function of $d$ when $m > 0$. Therefore, the max-path rule is expected to remain a useful heuristic even in the class-incremental setting.
>
> Since our similarity estimation method is not directly applicable to the class-incremental setting, applying this theoretical framework to nonlinear class-incremental image classification tasks remains a non-trivial challenge. Nonetheless, we believe this is an important direction for future work.
>
> > I think CIFAR-10/100 is still relatively simple and well-crafted, rather than "real-world".
>
> We agree that applying our method to datasets that better reflect real-world scenarios is an important direction for future work.

---

> > ### Comment · Reviewer_X6mc · 2025-04-05
> >
> > I thank the authors for their rebuttal. It addressed some of my concerns. I think the concerns about the use of task order has been partially addressed, where the proposed insight is applicable to the scenario with a small amount of available data rather than that without any multi-task data. As also mentioned by other reviewers, the real-world applicability remains a challenge. Overall, I appreciate the theoretical efforts and the interesting findings in this paper. I think it's valuable to better understand the continual learning problem. Therefore, I keep my rating toward accepting this paper.

---

### Official Review · Reviewer_Z3Hz · 2025-03-10

**Overall Recommendation:** 3

**Summary:**

This paper investigates the task order problem in continual learning. Specifically, it first establishes a theoretical connection between task order and final performance, which is then empirically validated through toy examples. Additionally, the paper examines the impact of task order from two key perspectives: task similarity and typicality. The findings suggest that an optimal task order should adhere to two principles: (1) tasks should be arranged from least representative to most typical, and (2) adjacent tasks should be more dissimilar. Experimental results on small-scale datasets demonstrate the effectiveness of the proposed method.

--------------- Post-Rebuttal Summary ---------------
========================
I appreciate the authors' efforts in preparing the rebuttal. While some of my concerns have been addressed, a few still remain. I will maintain my current rating as leaning toward acceptance. However, I strongly encourage the authors to include a discussion on the applicability of task order in real-world scenarios.

**Claims And Evidence:**

All claims are supported by some theoretical or empirical evidences.

**Essential References Not Discussed:**

No essential references are missed.

**Experimental Designs Or Analyses:**

Yes, I have reviewed the experimental designs and analyses in this paper. While I did not find any major issues, I have concerns regarding the experimental details and task scale.

**Methods And Evaluation Criteria:**

Yes, the proposed method and the adopted evaluation criteria align with the targeted problem and motivations.

**Other Comments Or Suggestions:**

Please refer to the **Other Strengths And Weaknesses**.

**Other Strengths And Weaknesses:**

Additional strengths:

 - This paper is well-motivated by both theoretical and empirical evidence, and the derived conclusions are clear and insightful.

- This paper is well organized.

Additional weaknesses:

- Does this imply that we have access to all tasks when adjusting task order? If so, why not use multi-task learning instead of continual learning?

- The evaluations are small-scale with limited comparisons. It remains unclear whether the task order problem is equally important across different CL approaches.

- All analyses are based on linear models and toy tasks (e.g., linear regression), limiting their generalizability.

- The methodology behind Figure 2(c) is unclear. Could the authors provide more experimental details on how this analysis was conducted?

- Some symbols lack explanation, e.g., $\delta$ in Line 160.

**Questions For Authors:**

Please refer to the **Other Strengths And Weaknesses**.

**Relation To Broader Scientific Literature:**

This paper have properly discussed with its counterparts.

**Theoretical Claims:**

I have generally checked proofs provided in the appendix.

---

> ### Author Rebuttal · Authors · 2025-03-31
>
> Thank you for your helpful comments. Please find our replies below (we quoted your comments only partially due to space constraint).
>
> > Does this imply that we have access to all tasks when adjusting task order?
>
> While we need some information on all tasks, we do not need to have access to all data for adjusting the task order.
> To clarity this point, we newly implemented an experiment where we estimated the optimal task order from a small subset of training data.
>
> | Data Size  | 0.1\%         | 1\%           | 10\%          | 100\%         |
> |:--:|:--:|:--:|:--:|:--:|
> | Peri-core  | 60.89 ± 0.94 | 63.30 ± 0.76 | 64.06 ± 0.71 | 64.55 ± 0.71 |
> | Max-path  | 59.75 ± 0.82 | 61.58 ± 0.81 | 62.07 ± 0.78 | 62.22 ± 0.73 |
> | Min-path  | 57.89 ± 0.82 | 56.19 ± 0.67 | 55.80 ± 0.67 | 55.28 ± 0.63 |
> | Core-peri  | 57.60 ± 1.15 | 52.65 ± 1.05 | 52.03 ± 0.97 | 51.02 ± 0.95 |
> | Control | *58.78 ± 0.62*  |
>
> The table above shows the average performance of various task orders on split CIFAR-10, where task similarity and the resulting order are estimated from a small subset of data.
> The "Data Size" in the top row indicates the percentage of training data used for the similarity evaluation described in Eq. 16. For example, when Data Size = 1\%, we used 1\% of one task’s training data to train a network and evaluated its zero-shot transfer performance using 1\% of training data from another task.
> The 100\% column corresponds to the task setting studied in the manuscript.
> As shown in the table, even when using only 1\% of the data, the max-path and periphery-to-core rules consistently outperformed the control, defined as the average performance over random task orders. Note that the control performance doesn't depend on the data size by definition.
>
> The results above suggest that task-order optimization is feasible even with limited observations, opening the door to many practical applications.
> For example, consider training a self-driving algorithm across three distinct environments: a suburban area in Phoenix, the city center of Tokyo, and the mountainous regions of the Swiss Alps. Due to geographical and economic constraints, it may be impractical to conduct simultaneous multi-task training across all three locations. While offline multi-task learning is an option, it requires full data acquisition beforehand. An alternative approach is to run small-scale pilot experiments in each environment, measure task similarity, and then determine the most effective order for conducting full-scale training.  Our work offers a theoretical foundation for this approach.
>
> A similar challenge arises in the context of multi-task learning in robotics. In real-world settings, switching between tasks often involves physically rearranging objects around the robot, which is both time-consuming and labor-intensive. As a result, switching tasks on a trial-by-trial basis is often impractical, necessitating block-wise training instead. In such scenarios, optimizing task order could help maximize average performance across all tasks.
>
> We therefore believe that task-order optimization with partial observations has strong potential for real-world application.
>
> > The evaluations are small-scale with limited comparisons...
>
> This is indeed an important direction for the future work.
>
> > All analyses are based on linear models and toy tasks (e.g., linear regression), limiting their generalizability.
>
> We agree that our theoretical results derived in a linear model have limited direct applicability. However, our numerical results indicate that the proposed ordering rules generalize to deep nonlinear neural networks, including convolutional networks and multi-layer perceptrons.
>
> > The methodology behind Figure 2(c) is unclear...
>
> In this figure, we parameterized three tasks $A$, $B$, and $C$ by their pairwise input correlations $(\rho_{AB}, \rho_{BC}, \rho_{CA})$ and identified the optimal task order for each triplet. For every combination of $(\rho_{AB}, \rho_{BC}, \rho_{CA})$, we computed the error for all six possible task orders using Eq. 8, and assigned a color to the point based on the order that yielded the lowest error. The optimal order was unique except along the boundaries between regions. We sampled $\rho_{AB}$ and $\rho_{BC}$ from the range $[0, 0.005, 0.01, \dots, 0.995]$, and $\rho_{CA}$ from $[0.2, 0.4, 0.6, 0.8]$, resulting in 160,000 evaluated triplets (which was made possible by our simple analytical expression). The white region corresponds to combinations of $(\rho_{AB}, \rho_{BC}, \rho_{CA})$ that do not constitute a valid correlation matrix. For example, if $\rho_{AB} = 1.0$ and $\rho_{BC} = 0.0$, then $\rho_{CA} = 0.8$ is not a valid value due to the constraints of correlation matrices.
>
> > Some symbols lack explanation, e.g., $\delta$ in Line 160.
>
> Thank you for pointing out. $\delta_{ik}$ in Line 160 is Kronecker delta, which returns 1 if $i = k$ and returns 0 if $i \neq k$.

---

> > ### Comment · Reviewer_Z3Hz · 2025-04-02
> >
> > I appreciate the authors' rebuttal, which has addressed some of my concerns. After reviewing their response and considering feedback from other reviewers, I would like to respond as follows:
> >
> > I acknowledge that MTL may not always be applicable. However, in the current version of this paper, it appears that all task data are accessible. Under such a setting, MTL is applicable or can at least serve as an upper bound. To better highlight the importance of task order, the authors should focus on optimizing task order with a subset of tasks, and this would require significant revisions to the writing.

---

### Official Review · Reviewer_r3vn · 2025-03-10

**Overall Recommendation:** 3

**Summary:**

This paper investigates the impact of task order on learning performance in continual learning. It first mathematically models the correlations between tasks and then derives two rules to enhance learning effectiveness: (1) learning non-representative tasks before typical ones, referred to as the *periphery-to-core rule*; and (2) ensuring that consecutive tasks are as dissimilar as possible, known as the *max-path rule*. Experimental results validate the effectiveness of these principles.

### Update after rebuttal
In summary, this paper has certain limitations in real-world applications. Maybe it is useful in some cases (e.g., the self-driving one you mentioned), but it is less practical than the existing setting of continual learning. However, I think this paper is a good and insightful attempt, so I will keep my initial recommendation, leaning towards accepting this paper.

**Claims And Evidence:**

Based on my understanding, the claims in this paper are well-founded and supported by rigorous reasoning.

**Essential References Not Discussed:**

I did not observe such cases.

**Experimental Designs Or Analyses:**

The paper presents sufficient experimental results with relatively thorough analysis. However, it lacks a clear table summarizing the performance impact of applying or not applying the two proposed rules.

**Methods And Evaluation Criteria:**

From a research perspective, the proposed method is innovative and carries some significance. However, I have concerns about its practical applicability. In real-world continual learning scenarios, task order is typically not controllable on demand. Continual learning simulates the process of continuously updating a model in response to updating environment. Since these changes are unpredictable, the learning order cannot be predetermined. As a result, the practical relevance of this work may be questionable.

**Other Comments Or Suggestions:**

In the third paragraph of Sec. 1, the authors provide several examples to illustrate scenarios where, even when all tasks are known in advance, it is necessary to learn only a subset at a time to reduce learning difficulty/computational cost. This argument appears to counter my concerns about the method’s limited practicality. However, I find the authors’ reasoning somewhat unconvincing.


In scenarios such as LLM training and robotics, while data may be processed in steps, each step typically follows a similar distribution. This differs from the continual learning setting, where each step introduces a distinctly different task. Using the authors’ example of Task A, Task B, and Task C, these scenarios are more akin to training on one-third of A’s data, one-third of B’s data, and one-third of C’s data simultaneously at each step, rather than sequentially learning A, then B, then C. As a result, these scenarios do not inherently involve issues like catastrophic forgetting, which continual learning methods aim to address, making the proposed approach less relevant.


The real risk of catastrophic forgetting arises in scenarios where future tasks are unknown in advance. However, the proposed method does not apply to such cases. Therefore, I would like the authors to clarify whether there exist application scenarios where (1) all tasks are fully known beforehand, yet (2) it is still necessary to learn only one task or a subset of tasks at a time.

**Other Strengths And Weaknesses:**

The strengths of this paper lie in its rigorous derivations and its novel perspective on continual learning tasks through task ordering. However, as noted in the **Methods and Evaluation Criteria** section, I have some doubts regarding its practical utility. Additionally, the paper lacks a clear table presenting intuitive results to demonstrate the effectiveness of the proposed method.

**Questions For Authors:**

I don’t have other questions.

**Relation To Broader Scientific Literature:**

The contributions of this paper are primarily within the field of continual learning. It may also be applicable to other continual learning scenarios beyond image classification.

**Theoretical Claims:**

Based on my understanding, I did not find any theoretical claims that appear to be obviously problematic.

---

> ### Author Rebuttal · Authors · 2025-03-31
>
> Thank you for your insightful comments. Please find our replies below.
>
> > it lacks a clear table summarizing the performance impact of applying or not applying the two proposed rules.
>
> The table below summarizes the results from Figs. 5a-f.
>
> |   | Fashion MNIST| CIFAR-10     | CIFAR-100    |
> |:--:|:--:|:--:|:--:|
> | Peri-core (\%)  | 67.25 ± 0.98 | 64.35 ± 0.54 | 63.96 ± 0.52 |
> | Max-path (\%)     | 66.03 ± 0.88 | 62.20 ± 0.56 | 62.61 ± 0.48 |
> | Min-path (\%)     | 56.96 ± 0.81 | 55.55 ± 0.51 | 57.50 ± 0.40 |
> | Core-peri (\%)  | 51.32 ± 1.16 | 51.15 ± 0.68 | 55.02 ± 0.57 |
> | Control (\%)    | 61.97 ± 0.78 | 59.10 ± 0.47 | 60.16 ± 0.38 |
>
> Here, we took the average over 100 random seeds, with errors representing the standard error of the mean (we crunched the x-axes of Fig. 5). This table confirms the robust advantage of the proposed rules.
> In Fig. 5, we depicted it in graph, as we identified mean task similarity, used as the x-axis, as a significant covariate.
> We will include the table in the revised manuscript.
>
> > In scenarios such as LLM training and robotics, while data may be processed in steps, each step typically follows a similar distribution.
>
> We believe our framework remains applicable even when tasks share similar distributions, as you may see in LLM training or robotics.
>
> To illustrate this, we applied our algorithms to Fashion-MNIST, using a random split of the training dataset into five sub-tasks. Specifically, we selected 500 images from the training set and created five sub-tasks, each containing 100 randomly chosen images. We considered a 10-class classification task here, as each sub-task included all 10 labels.
> We estimated the similarity by measuring the zero-shot transfer performance between sub-tasks using Eq. 16. Since the data was randomly split, all sub-tasks were inherently similar, but it varied across them.
> We then trained a network sequentially on the five sub-tasks using the inferred task orders. We found that task orders determined by the periphery-to-core rule and the max-path rule yielded better test accuracy than the control (see the table below). We observed the same trend with CIFAR-10, although overall performance was low in this low-sample regime.
>
> |   | Fashion MNIST |  CIFAR-10  |
> |:--:|:--:|:--:|
> | Peri-core (\%)  | 77.25 ± 0.19 | 31.70 ± 0.21 |
> | Max path (\%)   | 77.03 ± 0.13 | 31.67 ± 0.18 |
> | Min path (\%)   | 76.35 ± 0.15 | 30.64 ± 0.18 |
> | Core-peri (\%)  | 76.26 ± 0.25 | 30.24 ± 0.27 |
> | Control (\%)    | 76.63 ± 0.10 | 31.01 ± 0.13 |
>
> It remains unclear whether it scales to LLMs, where both the corpus size and input space are vast (due to the sequential nature of the input). Nevertheless, it highlights the potential utility of task-order optimization even when tasks follow highly similar distributions.
>
> > I would like the authors to clarify whether there exist application scenarios where (1) all tasks are fully known beforehand, yet (2) it is still necessary to learn only one task or a subset of tasks at a time.
>
> Regarding (1), please note that our framework does not require access to the full dataset for all tasks in advance.
> To clarify this point, we newly conducted an experiment where we used only a small subset of training data for estimating the task similarity and the optimal order (here we considered five binary classification tasks as in Fig. 5).
> The table attached to the reply to Reviewer Z3Hz describes the average performance of various task orders on split CIFAR-10 in this task setting. We found that even when using only 1\% of the data, the max-path and periphery-to-core rules consistently outperformed the random task order (please see the explanation below the table for details).
>
> This sufficiency of partial observation opens up many applications, because it enables a researcher to first collect a small amount of data for all tasks, then perform online learning using the inferred task order while gradually acquiring the full dataset.
> For example, consider training a self-driving algorithm across three distinct environments.
> Due to geographical and economic constraints, it may be impractical to conduct simultaneous multi-task training across all three locations.
> In such cases, a practical strategy is to run small-scale pilot experiments in each environment, measure task similarity, and determine the most effective order for full-scale training. Online learning can then proceed sequentially in each environment, without the need to wait for complete data collection from all locations.
> A similar strategy could be effective in medical image analysis, where data might need to be collected across different regions and diseases one by one.
>
> Regarding (2), in robotics, switching between tasks often involves physically rearranging objects around the robot, which is both time-consuming and labor-intensive. Thus, switching tasks on a trial-by-trial basis may be impractical, necessitating training on each task one at a time in a block-wise manner.

---

> > ### Comment · Reviewer_r3vn · 2025-04-03
> >
> > Thank you for your careful response. After reading your rebuttal, I have no more questions.
> >
> > In summary, this paper has certain limitations in real-world applications. Maybe it is useful in some cases (e.g., the self-driving one you mentioned), but it is less practical than the existing setting of continual learning. However, I think this paper is a good and insightful attempt, so I will keep my initial recommendation, leaning towards accepting this paper.

---

### Official Review · Reviewer_Z4Kz · 2025-03-16

**Overall Recommendation:** 2

**Summary:**

The paper considers task orders in continual learning. It formulates a linear teacher-student model and derives an explicit expression for the errors in the asymptotic case. Then the paper analyzes how the change of task order would affect the error.

**Claims And Evidence:**

Some claims made in the submission are not supported by clear and convincing evidence.

For example, the title is "Optimal Task Order for Continual Learning of Multiple Tasks". This is an overclaim, as the paper has not proved any optimality regarding task orders. The lack of serious proofs invalidates the claim that "this paper proposes two principles for task order optimization". The use of various heuristics under conveniently simplified settings is a contradiction to the word, "principle".

The paper also claims that it proposes two task ordering rules, the periphery-to-core rule and the max-path rule. This is also an overclaim, which the authors themselves actually know and cited: The max-path rule was considered in prior work of Bell & Lawrence (2022). Indeed, many experimental findings and figures follow from Bell & Lawrence (2022) in spirit.

**Essential References Not Discussed:**

I am not sure.

**Experimental Designs Or Analyses:**

Yes.

**Methods And Evaluation Criteria:**

The proposed formulation under the linear teacher-student model does not make too much sense (see comments below).

**Other Comments Or Suggestions:**

- The paragraph below Eq. 4 seems to have some formatting issues.
- At Eq 5, since the current task is solved optimally, why should one consider gradient descent specifically? gradient descent couldn't find the exact solution (5) unless it is run in infinitely many iterations
- Line 195 (left): permutation-invatiant
- Line 188 (right): Grammar at sentence "If $\rho\_{AB}=1$ ..."
- Line 271 (left): "mroe"
- Section 4.3 should point out that finding the longest Hamiltonian path in a (weighted) graph is NP-hard.
- The description "periphery-to-core" is very confusing, and I found Figure 3c not useful. The tasks defined as per the paper have no "spatial" relations. In the caption, why tasks A-C are central and D-I are periphery? Is that because they are plotted that way?
- Line 423 (left): "bu"

**Other Strengths And Weaknesses:**

## Linear teacher-student models
- The paper advocates using a linear teacher-student model. However, this viewpoint is questionable, as it only introduces notational complexity and never provides new insights. Variables $s, x, y, y^*$ later never appear in the paper. At the end of the day, the objective the paper aims to minimize is (3), which is a least-squares objective. From this viewpoint, the setting has little difference from prior work focusing on linear models (e.g. Lin et al 2023). By directly introducing objective (3), the paper would be much more readable.
- The distribution assumption such as (2) is not very well motivated. Why would the data of different tasks be related in such a specific way? Practical examples are needed to justify this assumption.



## Clarity and Motivation
The clarity of the paper is largely compromised due to the use of non-standard notations. For example:
- The paper uses the symbol of inner product to denote expectation (see Eq 3). Then, also in Eq 3, $\epsilon$ is defined as an error function of weight $W$, and $\overline{\epsilon}$ is defined as the minimum average error, but this is now a function of correlation matrices. But later in Theorem 4.1, $\overline{\epsilon}[m,\rho\_o]$ seems to suggest that $\overline{\epsilon}$ is a function of two numbers, $m$ and $\rho\_o$.
- The symbol $\delta$ was used at Line 160, and its meaning is not defined. This is confused with $\delta M$, which means the perturbation. The confusion deepens when we see $G_{\mu \nu} \delta M_{\mu \nu}$ in Theorem 4.1, while Theorem 4.1 contains another $\delta$, which is $\delta_m$, some positive constant.

At various places, the definitions of certain quantities lack motivations as well. For example:
- While it is understandable that finding an optimal task order is difficult, the introduction of linear perturbation theory needs a better motivation. Why could a small perturbation indicate a small final error? If this issue is not addressed, going ahead and optimizing $G_{\mu \nu}$ appears to be very heuristic.
- At Line 220 (right), why does $\delta t_{\mu}$ defined in such a way indicate "relative typically of the task"?
- At line 267 (right), why does $d_{\mu \nu}$ defined in such as way indicate "dissimilarity"?

**Questions For Authors:**

See the above.

**Relation To Broader Scientific Literature:**

I am not sure.

**Theoretical Claims:**

I didn't check the correctness of the proofs.

---

> ### Author Rebuttal · Authors · 2025-03-31
>
> Thank you for your detailed comments on our work. Please find the replies below (we quoted your comments only partially due to space constraint).
>
> **Claims And Evidence**
>
> > For example, the title is...
>
> Please note that, both in the abstract and main text, we noted that these principles are based on linear models and apply only to specific parameter ranges.
>
> > The paper also claims that it proposes two task ordering rules,...
>
> While Bell \& Lawrence (2022) is an important inspiration, our work represents a significant advancement beyond theirs.
> First, we develop a theoretical framework to explain how task similarity and task order impact continual learning performance, in contrast to their primarily empirical approach. Specifically, our analysis sheds light on why the max-path on a task dissimilarity graph often improves performance and how the number of tasks influences this trend.
> In addition, we introduce and analyze an alternative heuristic for task-order optimization: the periphery-to-core rule.
> Finally, unlike Bell \& Lawrence (2022), our task similarity inference method does not require access to model parameters.
>
> **Other Strengths And Weaknesses**
>
> > The paper advocates using a linear teacher-student model. However,...
>
> The variables $\boldsymbol{x}$, $\boldsymbol{y}$, and $\boldsymbol{y^*}$ represent the input, output, and target, and are essential for defining the problem. The latent factor $\boldsymbol{s}$ allows flexible control over task similarity in the input and output spaces ($C^{in}$ and $C^{out}$). This setup enables us to examine how input-space similarity, $C^{in}$, influences continual learning performance and optimal task order, which previous linear models (e.g., Lin et al., 2023) do not capture.
>
> > The distribution assumption such as (2) is not very well motivated.
>
> This specific assumption was introduced for analytical tractability. However, we later showed that task similarity can be estimated from data, and its effects on task ordering and continual learning performance are consistent with our theory.
>
> > The paper uses the symbol of inner product to denote expectation (see Eq 3).
>
> $\langle x \rangle$ symbol is commonly used to denote the expectation of $x$ in the field of mathematical sciences and machine learning works employing mathematical physics methodology.
>
> > Then, also in Eq 3, $\epsilon$ is defined as ...
>
> We defined the error on task $\mu$, $\epsilon_{\mu}$, as a function of the model weights. We then derived the average error, $\bar{\epsilon}_f$, in terms of the task correlation matrices to analyze how task correlations impact performance. In Theorem 4.1, we focused on uniform correlation matrices parameterized by $m$ and $\rho_o$, up to a small perturbation term, and denoted the corresponding error as $\bar{\epsilon}_f[m, \rho_o]$.
>
> > The symbol $\delta$ was used at Line 160,...
>
> Thank you for pointing out. The definition of delta was indeed missing at L160. Here, $\delta_{ik}$ is defined as the Kronecker delta, which returns 1 if $i = k$, and returns 0 otherwise.
> $\delta M$ is a matrix, as noted in Theorem 4.1, and $\delta M_{\mu\nu}$ is the $(\mu,\nu)$-th element of matrix $\delta M$. $\delta_m$ is a small positive constant defined as the upper bound of $\delta M_{\mu\nu}$. We will clarify these points in the revision.
>
> > While it is understandable that finding an optimal task order is difficult,...
>
> A small perturbation does not imply a small final error; rather, it implies a small change in the final error. This allows us to use a first-order approximation to analyze how slight changes in task similarity affect the error.
> Furthermore, we did not optimize $G_{\mu\nu}$. This coefficient quantifies how a small change in the similarity between the $\mu$-th and $\nu$-th tasks influences the final error.
>
> > At Line 220 (right),
>
> $\delta M_{\mu\nu}$ captures the relative similarity between tasks $\mu$ and $\nu$, thus its summation $\sum_{\nu\neq\mu} \delta M_{\mu\nu}$ represents how similar task $\mu$ is with all other tasks, which in turn measures the typicality of task $\mu$.
>
> > At line 267 (right),...
>
> $C_{\mu\nu}^{in}$ captures the correlation between tasks $\mu$ and $\nu$ in the input space (see Eq. 2), and thus $d_{\mu\nu} \equiv 1 - C_{\mu\nu}^{in}$ represents the dissimilarity.
>
> **Other Comments Or Suggestions**
>
> > At Eq 5,
>
> In our setting, there are infinitely many solutions that minimize task-specific error. We focus on the solution reached by gradient descent, as it is the standard optimization method for neural networks. While convergence technically requires infinite iterations, the convergence rate is fast due to convexity.
>
> > The description "periphery-to-core" is very confusing,...
>
> While Figs. 3b and c are illustrations, they portray the multi-dimensional scaling (MDS) visualization of the task similarity matrix. For instance, in the chain-like task similarity depicted in Fig. 4a, task C is central whereas tasks A and E are peripheral.

---

### Decision · Program_Chairs · 2025-05-01

**Decision:**

Accept (poster)

**Comment:**

[1]From a research perspective, the proposed method is innovative and carries some significance. [Reviewer r3vn]
[2] The paper presents sufficient experimental results with relatively thorough analysis [Reviewer r3vn, Z3Hz]
[3] 	This paper is well-motivated by both theoretical and empirical evidence, and the derived conclusions are clear and insightful [Reviewer Z3Hz, Reviewer X6mc]
[4] The strengths of this paper lie in its rigorous derivations and its novel perspective on continual learning tasks through task ordering. [Reviewer r3vn]

Weakness:
[1] Applicability of task order in real-world scenarios of continual learning is ambiguous [Reviewer r3vn, Reviewer Z3Hz, Reviewer X6mc]